 

# Optogenetic control of PRC1 reveals its role in chromosome alignment on the spindle by overlap length-dependent forces

**Mihaela Jagrić[†], Patrik Risteski[†], Jelena Martinčić[†], Ana Milas, Iva M Tolić***

Division of Molecular Biology, Ruđer Bošković Institute, Zagreb, Croatia

**Abstract** During metaphase, chromosome position at the spindle equator is regulated by the forces exerted by kinetochore microtubules and polar ejection forces. However, the role of forces arising from mechanical coupling of sister kinetochore fibers with bridging fibers in chromosome alignment is unknown. Here, we develop an optogenetic approach for acute removal of PRC1 to partially disassemble bridging fibers and show that they promote chromosome alignment. Tracking of the plus-end protein EB3 revealed longer antiparallel overlaps of bridging microtubules upon PRC1 removal, which was accompanied by misaligned and lagging kinetochores. Kif4A/kinesin-4 and Kif18A/kinesin-8 were found within the bridging fiber and largely lost upon PRC1 removal, suggesting that these proteins regulate the overlap length of bridging microtubules. We propose that PRC1-mediated crosslinking of bridging microtubules and recruitment of kinesins to the bridging fiber promote chromosome alignment by overlap length-dependent forces transmitted to the associated kinetochore fibers.

**\*For correspondence:**
tolic@irb.hr

[†]These authors contributed equally to this work

**Competing interests:** The authors declare that no competing interests exist.

## Introduction

Pre-anaphase chromosome movements culminate with chromosome alignment at the spindle equator, a distinctive feature of mitosis important for the synchronous anaphase poleward movement of chromatids and proper telophase nuclear reformation (*Fonseca et al., 2019*; *Maiato et al., 2017*). Chromosome congression to the metaphase plate, a process of directed chromosome movement from the polar regions toward the spindle equator, has been explored extensively (*Barisic et al., 2014*; *Cai et al., 2009*; *Kapoor et al., 2006*; *Maiato et al., 2017*). Yet, the maintenance of chromosome alignment at the spindle equator is less understood. This is a dynamic process given that the chromosomes constantly make small oscillatory excursions across the equator (*Skibbens et al., 1993*). Two mechanisms have been proposed to underlie the maintenance of chromosome alignment at the equator, namely length-dependent dynamics of kinetochore fiber (k-fiber) microtubules and polar ejection forces.

Pulling forces exerted by k-fiber tips on kinetochores are thought to be regulated by kinesin-8 motor proteins, which are needed for proper kinetochore alignment in various organisms from yeast to humans (*Garcia et al., 2002*; *Stumpff et al., 2008*; *Wargacki et al., 2010*). These motors can 'measure' microtubule length because they bind to microtubules along their length and move toward the plus end, leading to more kinesin-8 accumulated at the plus end of longer microtubules, where they promote microtubule catastrophe, that is, a switch from growth to shrinkage (*Tischer et al., 2009*; *Varga et al., 2006*). During kinetochore movements across the spindle equator, the leading k-fiber shrinks, while the trailing one grows and accumulates kinesin-8, resulting in microtubule catastrophe, followed by the movement of the trailing kinetochore back toward the equator. Although this mechanism can explain kinetochore alignment at the spindle equator

**eLife digest** Before cells divide to create copies of themselves, they need to duplicate their genetic material. To help split their DNA evenly, they build a machine called the mitotic spindle. The mitotic spindle is made of fine, tube-like structures called microtubules, which catch the chromosomes containing the genetic information and line them up at the center of the spindle.

Microtubules push and pull the chromosomes by elongating or shortening their tips. But it remains unclear how the microtubules know when the chromosomes have reached center point. One way to find out is to remove proteins that accumulate in the middle of the spindle during division, such as the protein PRC1, which helps to assemble a subset of microtubules called bridging fibers, and the proteins Kif4A and Kif18A, which work like molecular rulers, shortening long microtubules. Usually, scientists would delete one of these proteins to see what impact this has. However, these experiments take days, giving the cell enough time to adapt and thus making it difficult to study the role of each of the proteins.

Here, Jagrić, Risteski, Martinčić et al. used light to manipulate proteins at the exact moment of chromosome alignment and to move PRC1 from the spindle to the cell membrane. Consequently, Kif4A and Kif18A were removed from the spindle center. This caused the bridging fibers, which overlap with the microtubules that connect to the chromosomes, to become thinner. Jagrić et al. discovered that without the molecular ruler proteins, the bridging fibers were also too long. This increased the overlap between the microtubules in the center of the spindle, causing the chromosomes to migrate away from the center. This suggests that the alignment of chromosomes in the middle of the spindle depends on the bridging microtubules, which need to be of a certain length to effectively move and keep the chromosomes at the center. Thus, forces that move the chromosomes are generated both at the tips of the microtubules and along the wall of microtubules.

These results might inspire other researchers to reassess the role of bridging fibers in cell division. The optogenetic technique described here could also help to determine the parts other proteins have to play. Ultimately, this might allow researchers to identify all the proteins needed to align the chromosomes.

(*Klemm et al., 2018*), the switching dynamics characteristic for this model, where the trailing kinetochore initiates the change of direction of motion, differs from the observations in mammalian cells where the leading kinetochore typically changes the direction before the trailing one (*Armond et al., 2015*; *Civelekoglu-Scholey et al., 2013*; *Wan et al., 2012*).

In addition to the forces produced by k-fibers, polar ejection forces push chromosome arms away from the pole, powered by arm-bound chromokinesins that walk toward the plus end of microtubules (*Bajer et al., 1982*; *Brouhard and Hunt, 2005*). Because microtubule density increases toward the pole, these forces help the chromosomes to stay away from the poles, but most likely have little effect on kinetochore movements close to the spindle equator (*Cane et al., 2013*; *Ke et al., 2009*). Thus, the current models do not provide a complete picture of kinetochore alignment at the spindle center.

K-fibers are surrounded by a dense network of spindle microtubules with which they have multiple interactions (*McDonald et al., 1992*; *O'Toole et al., 2020*), resulting in forces acting on k-fibers and thus also on kinetochores. In particular, each pair of sister k-fibers is tightly linked by the bridging fiber, a bundle of antiparallel microtubules that balances the tension on sister kinetochores (*Kajtez et al., 2016*; *Polak et al., 2017*; *Vukušić et al., 2017*). However, the role of forces exerted by bridging fiber in chromosome alignment at the metaphase plate is unknown.

In metaphase, overlap regions within bridging fibers are crosslinked by protein regulator of cytokinesis 1 (PRC1) (*Kajtez et al., 2016*; *Polak et al., 2017*; *Tolić, 2018*). PRC1, like other non-motor microtubule-associated proteins from Ase1/PRC1/MAP65 family, selectively bundles antiparallel microtubules and provides stable overlaps *in vitro* (*Bieling et al., 2010b*; *Janson et al., 2007*; *Mollinari et al., 2002*; *Subramanian et al., 2010*). Cellular studies of its function show that PRC1 is associated with the spindle midzone in anaphase, where its activity is essential for stable microtubule organization, localization of numerous microtubule-associated proteins within this structure, and

successful completion of cytokinesis, while its microtubule-binding and -bundling affinities are regulated by phosphorylation and dephosphorylation events (*Gruneberg et al., 2006*; *Jiang et al., 1998*; *Kurasawa et al., 2004*; *Liu et al., 2009*; *Mollinari et al., 2002*; *Mollinari et al., 2005*; *Neef et al., 2007*; *Subramanian et al., 2013*; *Subramanian et al., 2010*; *Zhu and Jiang, 2005*; *Zhu et al., 2006*).

In this work, we developed an optogenetic approach for acute and reversible removal of PRC1 from the spindle to the cell membrane, building upon ideas of dimerization or dissociation induced chemically (*Cheeseman et al., 2013*; *Haruki et al., 2008*; *Robinson et al., 2010*; *Wordeman et al., 2016*) or by light (*Fielmich et al., 2018*; *Guntas et al., 2015*; *Okumura et al., 2018*; *van Haren et al., 2018*; *Yang et al., 2013*; *Zhang et al., 2017*) to rapidly redistribute proteins. By using our assay on metaphase spindles, we found that bridging fibers promote kinetochore alignment. PRC1 removal resulted in partial disassembly of bridging fibers and elongation of their overlap zones. Moreover, the metaphase plate widened, inter-kinetochore distance decreased, and lagging chromosomes appeared more frequently, showing that PRC1 indirectly regulates forces acting on kinetochores. Kif4A/kinesin-4 and Kif18A/kinesin-8 were found to localize in the bridging fiber during metaphase and were largely lost upon PRC1 removal, with Kif4A showing a greater reduction. These results, together with the finding that Kif4A or Kif18A depletion by siRNA led to elongated overlaps, suggest that these proteins regulate the overlap length of bridging microtubules. PRC1 removal did not affect the localization of Kif4A on the chromosomes and Kif18A, CLASP1, and CENP-E/kinesin-7 on the plus ends of k-fibers, arguing against perturbed polar ejection forces or molecular events at the kinetochore microtubule plus ends as origins of the observed kinetochore misalignment. In conclusion, our optogenetic experiments show that bridging microtubules buffer chromosome movements, thus promoting their alignment. We propose that this occurs via length-dependent forces, which depend on the antiparallel overlap length within the bridging fiber.

## Results

### Optogenetic system for fast and reversible removal of PRC1 from the metaphase spindle

To study the role of PRC1 and the forces arising from coupling of bridging and k-fibers in chromosome alignment, we developed an optogenetic tool for fast and reversible removal of PRC1 from the spindle to the cell membrane, based on the previously designed improved light inducible dimer (iLID) system (*Guntas et al., 2015*). We attached PRC1 to the red fluorescent protein tgRFPt and the bacterial protein SspB, while the iLID, which contains the bacterial peptide SsrA and the light-oxygen-voltage (LOV2) domain, is bound to the cell membrane by a short peptide, CAAX. In this system, LOV2 adopts a conformation that allows dimerization of SsrA and SspB upon exposure to the blue light (*Figure 1A*). After cessation of exposure to the blue light, LOV2 adopts its initial conformation leading to decreased affinity of SsrA to SspB. Therefore, exposure to the blue light should induce translocation of PRC1 from the central region of the metaphase spindle, which we will refer to as the spindle midzone, to the cell membrane, whereas cessation of exposure to blue light should restore PRC1 localization on the spindle (*Figure 1A*).

To test our optogenetic approach, we used U2OS cells with stable expression of CENP-A-GFP, transient expression of PRC1-tgRFPt-SspB (henceforth opto-PRC1) and iLID-CAAX (henceforth opto cells; *Figure 1B*; *Figure 1—video 1*). Endogenous PRC1 was depleted 90 ± 2% (all results are mean ± s.e.m.) by siRNA before addition of opto-PRC1 (*Figure 1—figure supplement 1A*). Before exposure to the blue light, opto-PRC1 had normal localization on the microtubule bundles in the spindle midzone (*Figure 1B*; 0:00 min), with the length of PRC1 streaks of 3.77 ± 0.08 μm (n = 193 bundles, N = 30 cells), consistent with that of endogenous and fluorescently labeled PRC1 in metaphase (*Kajtez et al., 2016*; *Polak et al., 2017*), although the total signal intensity of opto-PRC1 on the spindle was higher compared to endogenous PRC1 (*Figure 1—figure supplement 1B*). Addition of opto-PRC1 did not change the duration of metaphase, as inferred from the fraction of cells that entered anaphase during image acquisition, which was similar in cells with endogenous PRC1 and cells treated with PRC1 siRNA and containing opto-PRC1 (79 ± 6%, N = 37, and 71 ± 5%, N = 72, respectively; p = 0.4, Pearson's Chi-squared test; *Figure 1—figure supplement 1C*). After exposure

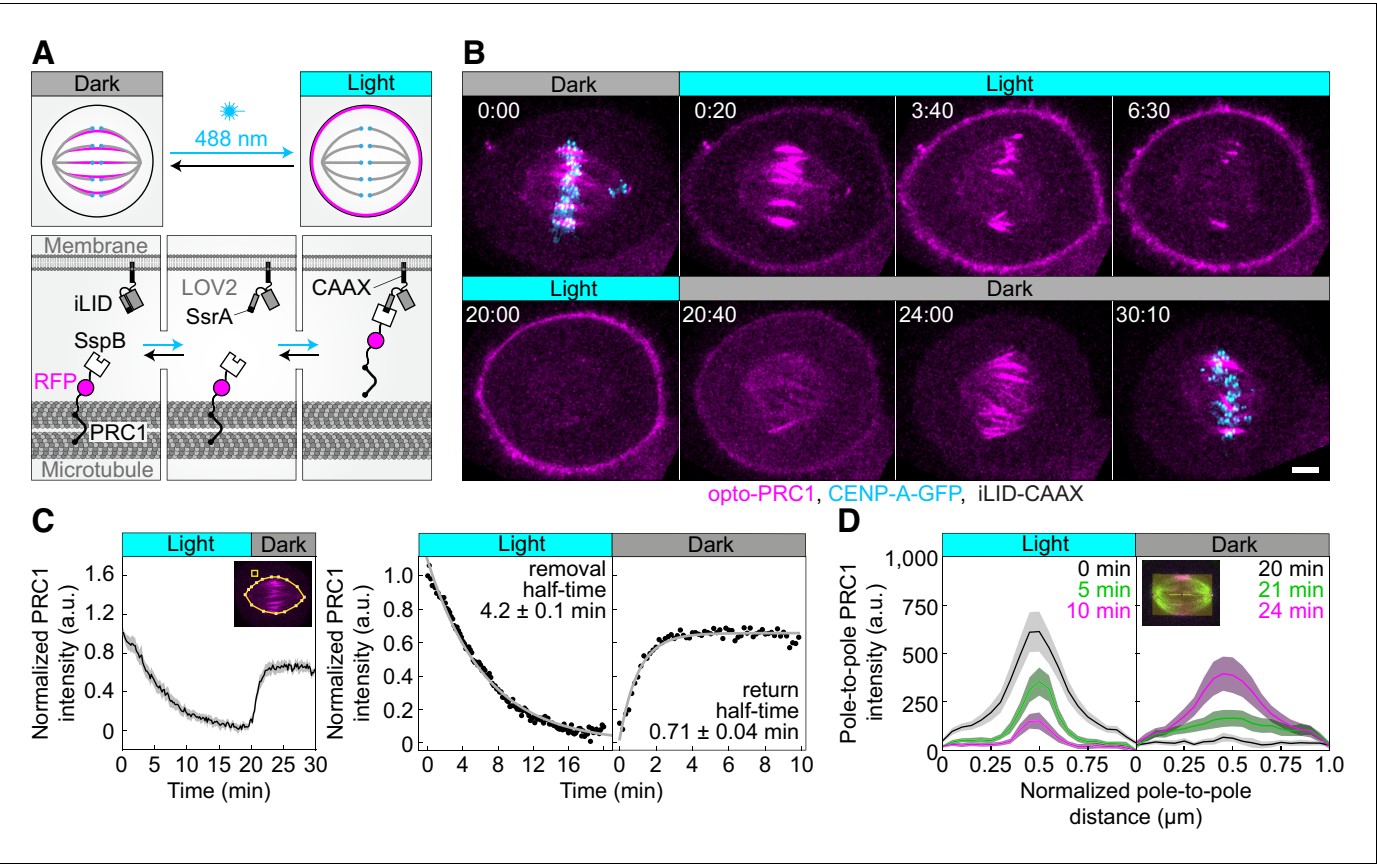

**Figure 1.** Optogenetic reversible removal of PRC1 from the spindle in metaphase. (**A**) Schematic representation of the optogenetic system. PRC1 is fused with SspB and tgRFPt (opto-PRC1, see Materials and methods). iLID, composed of photosensitive LOV2 domain and SsrA is tagged with CAAX sequence which mediates its binding to the cell membrane. Exposure to the blue light induces conformational change in LOV2 domain, enabling dimerization of SspB and SsrA, and thus translocation of PRC1 from the spindle to the cell membrane. After the blue light is turned off, LOV2 adopts its initial conformation, leading to decreased affinity of SspB for SsrA, and consequently dissociation of PRC1 from the membrane and its return to the spindle. (**B**) Time-lapse images of a metaphase spindle in a U2OS cell stably expressing CENP-A-GFP (cyan), depleted for endogenous PRC1, with transient expression of opto-PRC1 (magenta) and iLID-CAAX. Note that kinetochores are shown only in the first and the last time frame in order to better visualize PRC1 removal. Images are maximum intensity projections of three z-planes, smoothed with 0.1-μm-sigma Gaussian blur. Time: min:s. Scale bar; 5 μm. (**C**) Normalized intensity of opto-PRC1 signal on the spindle (left panel) during its removal (0–20 min) and return (20–30 min). N = 15 (see *Figure 1—figure supplement 1F* for individual cells). Scheme depicts the areas where opto-PRC1 intensity was measured: spindle (large polygon) and cytoplasm (small square). Exponential fit (gray lines in the right panel) on mean normalized opto-PRC1 spindle intensity (black points) during 20 min of removal and 10 min of return. Formulae y=A*exp(-τ*x) and y=A*exp(-τ*x)+c were used for opto-PRC1 removal and return, respectively. Parameters for PRC1 removal: A = 1.111, τ = 0.00277 s$^{-1}$ (RSE = 0.03), and return: A = −0.635, c = 0.656, τ = 0.01622 s$^{-1}$ (RSE = 0.03). The half-time was calculated by ln2/τ. (**D**) Pole-to-pole opto-PRC1 intensity during removal (left; N = 14) and return (right; N = 9) to the spindle. Mean and s.e.m are color-coded corresponding to the time when measured (upper right corners). Scheme depicts the area where opto-PRC1 intensity was measured (yellow) to obtain the mean intensity projection onto the pole-to-pole axis. Mean (thick lines); s.e.m. (shaded areas); N (number of cells).

The online version of this article includes the following video and figure supplement(s) for figure 1:

**Figure supplement 1.** Validation of optogenetic system and PRC1 removal.

**Figure 1—video 1.** U2OS cell stably expressing CENP-A-GFP (cyan), with transient expression of opto-PRC1 (magenta) and iLID-CAAX.
https://elifesciences.org/articles/61170#fig1video1

to the blue light, opto cells were able to progress to cytokinesis (*Figure 1—figure supplement 1D*). Taken together, these data suggest that opto-PRC1 replaces the depleted endogenous PRC1.

Upon exposure to the blue light, opto-PRC1 signal on the spindle decreased and its signal on the membrane increased (*Figure 1B*; 0:20-20:00 min). After the blue light was switched off, opto-PRC1 returned to the spindle midzone (*Figure 1B*; 20:40-30:10 min). To validate our system, we performed two sets of test experiments. The first one was performed on the same cell line containing opto-PRC1, but without iLID, imaged with the same imaging protocol as the opto cells, which we refer to

as control throughout the paper. The second test experiment was on the opto cells but without the blue light (*Figure 1—figure supplement 1E*). In both cases, opto-PRC1 remained on the spindle (*Figure 1—figure supplement 1E*). Thus, our optogenetic approach allows for acute and reversible control of PRC1 localization in metaphase.

To quantify the dynamics and spatial pattern of opto-PRC1 removal and return, we measured the intensity of opto-PRC1 on the metaphase spindle (*Figure 1—figure supplement 1F,G*). We found that $88 \pm 3\%$ of opto-PRC1 was removed after 20 min of exposure to the blue light with a half-time of $4.2 \pm 0.1$ min (*Figure 1C*). During the opto-PRC1 removal, there was simultaneous decrease in both signal intensity and length of the overlap region (*Figure 1D*, left; *Figure 1—figure supplement 1G*, top), which may be due to fewer antiparallel regions being positioned laterally than in the central part of the bundle (*Mastronarde et al., 1993*). The signal of the outermost midzone bundles typically lasted longer than of the inner ones (*Figure 1B*; 3:40-6:30). After the blue light was switched off, opto-PRC1 signal restored to $65 \pm 1\%$ of the initial intensity within 10 min, with return half-time being $0.71 \pm 0.04$ min (*Figure 1C*). The faster PRC1 return to the spindle in comparison with its removal may be due to the higher affinity difference between PRC1 binding to the spindle and to the membrane in the dark than under light. During the opto-PRC1 return, it initially localized throughout the spindle, with gradual increase in intensity in the spindle midzone (*Figure 1D*, right; *Figure 1—figure supplement 1G*, bottom), suggesting that PRC1 has higher unbinding rate outside than within the overlap bundles in the spindle. This result is consistent with PRC1 having a life-time of several seconds on single microtubules and a 10-fold preference for overlap regions *in vitro* (*Subramanian et al., 2010*).

## Acute PRC1 removal during metaphase leads to misaligned kinetochores

To test whether bridging fiber has a role in the maintenance of chromosome alignment, acute optogenetic removal of PRC1, a depletion of which is known to perturb bridging fibers (*Polak et al., 2017*) should affect chromosome positioning at the spindle equator (*Figure 2A*). Surprisingly, we observed that the acute removal of opto-PRC1 resulted in movement of sister kinetochore pairs away from the metaphase plate (*Figure 2B*; *Figure 2—video 1*), which is not found after long-term PRC1 depletion by siRNA (*Polak et al., 2017*). Upon opto-PRC1 removal, the distances of sister kinetochore midpoints from the equatorial plane increased ($d_{EQ}$, *Figure 2B,C,D*; *Figure 2—figure supplement 1A,B*). While >95% of kinetochore pairs were found within the region of PRC1 streaks, that is, less than 2 µm away from the equatorial plane before removal of opto-PRC1, $9.3 \pm 2.7\%$ of kinetochore pairs made excursions far outside this region after opto-PRC1 removal (*Figure 2—figure supplement 1B*). The displaced kinetochores were found more often in the inner part of the spindle, that is, close to the long spindle axis, than in the outer regions (*Figure 2E*). Kinetochores fluctuated to a similar extent in the presence and absence of opto-PRC1, but in its absence the displaced kinetochores fluctuated within a region that was offset from the equatorial plane (*Figure 2—figure supplement 1C*). These displaced kinetochores upon opto-PRC1 removal had lower inter-kinetochore distance in comparison to non-displaced ones, suggesting that kinetochore displacement was related to a more severe reduction of tension (*Figure 2F*). On average, kinetochores remained displaced even after opto-PRC1 return (*Figure 2D*). We conclude that PRC1 has a role in keeping kinetochores in tight alignment on the metaphase plate.

The mean inter-kinetochore distance ($d_{KC}$, *Figure 2C*) was reduced when opto-PRC1 was removed (*Figure 2G*; *Figure 2—figure supplement 1D*), and the distance after 20 min of exposure to the blue light ($0.79 \pm 0.01$ µm) was closer to metaphase ($0.87 \pm 0.01$ µm) than prometaphase ($0.66 \pm 0.01$ µm) values (see Materials and methods), suggesting that tension was not completely lost and that these changes were not due to kinetochore detachment from k-fibers (*Figure 2—figure supplement 1D*). In agreement with this, the fraction of cells that entered anaphase during image acquisition was similar in control and opto cells ($71 \pm 5\%$, N = 72, and $60 \pm 5\%$, N = 93, respectively; p = 0.1, Pearson's Chi-squared test; *Figure 1—figure supplement 1C*), indicating that PRC1 removal did not prevent spindle assembly checkpoint satisfaction. After opto-PRC1 return to the spindle, inter-kinetochore distance increased, implying restoration of tension, although not to the original value (*Figure 2G*).

To investigate the influence of acute PRC1 removal on the orientation of sister kinetochores, we measured the angle between sister kinetochore axis and long spindle axis ($\alpha_{KC}$, *Figure 2C*). We



**Figure 2.** Optogenetic removal of PRC1 disrupts kinetochore alignment on the metaphase plate and leads to lagging kinetochores in anaphase. (A) Schematic representation of possible outcomes of acute removal of PRC1 from the spindle on chromosome alignment. (B) Spindle in a U2OS cell stably expressing CENP-A-GFP (cyan) with transient expression of opto-PRC1 (magenta) and iLID-CAAX before (0 min, Dark), and at the end of continuous exposure to the blue light (18 min, right). Enlargements show kinetochores only. Scale bar; 2 μm. (C) Schematic of measured parameters. $d_{KC}$, inter-kinetochore distance; $d_{EQ}$, distance between sister kinetochore midpoint and equatorial plane (EQ); $\alpha_{KC}$, angle between sister kinetochore axis and spindle long axis (AX); $d_{AX}$, distance between sister kinetochore midpoint and spindle long axis. (D) Measurements of $d_{EQ}$ in opto (black), control (gray), and untreated (magenta) cells before (0 min, Dark), at the end of continuous exposure (20 min, Light) and 10 min after cessation of exposure to the blue light (30 min, Dark), in U2OS cells expressing CENP-A-GFP. (E) $d_{AX}$ of aligned ($d_{EQ} < 2$ μm) and misaligned ($d_{EQ} = 2.5 \pm 0.2$ μm) kinetochore pairs upon PRC1 removal. (F) $d_{KC}$ of aligned and misaligned kinetochore pairs upon PRC1 removal. (G) Measurements of $d_{KC}$. Legend as in D. (H) Measurements of $\alpha_{KC}$. Legend as in D. (I) Time-lapse images of a spindle in a U2OS cell as in B, stained with SiR-tubulin (not shown). Anaphase onset is at time 0 min. Lagging kinetochores can be seen at 3 min (middle). Enlargement shows kinetochores only. Scale bar: 5 μm. (J) Occurrence of lagging kinetochores in anaphase of opto (black) and control (gray) U2OS cells. (K) Schematic of three mechanisms that could be involved in kinetochore alignment. Cyan rectangles inside graphs indicate exposure to the blue light. Numbers in brackets denote measurements and cells; single numbers denote cells. In D, G, H, opto cells include those with and without SiR-tubulin, control cells without iLID are SiR-tubulin only, whilst untreated cells include only those without SiR-tubulin. Kinetochore parameters in control cells and untreated cells were similar; $d_{KC}$ p = 0.69; one-way ANOVA, $d_{EQ}$ p = 0.88, $\alpha_{KC}$ p = 0.27; Kruskal-Wallis rank sum test. All images are maximum intensity projections of three z-planes, smoothed with 0.5-pixel-sigma Gaussian blur. Error bars; s.e.m. Statistical analysis; Kruskal-Wallis rank sum test followed by pairwise Wilcoxon rank sum test (D, H), one-way ANOVA followed by

*Figure 2 continued on next page*

Figure 2 continued

Tukey Honest Significant Difference (HSD) post hoc test (**G**), t-test (**E, F**), two-proportions z-test (**J**). p-value legend:<0.0001 (****), 0.0001 to 0.001 (***), 0.001 to 0.01 (**), 0.01 to 0.05 (*),≥0.05 (ns).

The online version of this article includes the following video and figure supplement(s) for figure 2:

**Figure supplement 1.** Quantification of kinetochore misalignment, misorientation and tension upon PRC1 removal.

**Figure supplement 2.** Characterization of lagging kinetochores upon PRC1 removal.

**Figure 2—video 1.** U2OS cell stably expressing CENP-A-GFP (cyan), with transient expression of opto-PRC1 (magenta) and iLID-CAAX.

https://elifesciences.org/articles/61170#fig2video1

**Figure 2—video 2.** U2OS cell stably expressing CENP-A-GFP (cyan), with transient expression of opto-PRC1 (magenta) and iLID-CAAX, and microtubules stained with SiR-tubulin (green).

https://elifesciences.org/articles/61170#fig2video2

observed that removal of opto-PRC1 caused misorientation of sister kinetochores, that is, increased $\alpha_{KC}$ (**Figure 2B,H**; **Figure 2—figure supplement 1A,E**). Misoriented kinetochores were found at a larger distance from the equatorial plane and closer to the long spindle axis (**Figure 2—figure supplement 1F**). Interestingly, sister kinetochore pairs remained misoriented even after opto-PRC1 return (**Figure 2H**). Similarly, PRC1 bundles were misoriented upon PRC1 return (see Materials and methods, **Figure 2—figure supplement 1G**). These results suggest that when PRC1 returns to the overlaps whose geometry was perturbed by PRC1 removal, it likely confines the chromosomes in new positions and orientations.

The observed effects of PRC1 removal on inter-kinetochore distance, kinetochore alignment and orientation did not change when SiR-tubulin was added ($d_{KC}$ p = 0.82, $d_{EQ}$ p = 0.27, $\alpha_{KC}$ p = 0.61, respectively; t-test). The effects of PRC1 removal were found neither in control experiments in cells without iLID, which were stained with SiR-tubulin, nor in a different set of control experiments where cells expressing only CENP-A-GFP without SiR-tubulin were imaged with the same illumination protocol (**Figure 2D,G,H**; **Figure 2—figure supplement 1A,B,D,E**). Therefore, the observed effects in opto cells were not a consequence of SiR-tubulin or laser photodamage (**Douthwright and Sluder, 2017**).

As previous reports have shown that acute rapamycin-dependent protein translocation and long-term depletion by siRNA can yield different and even opposite phenotypes (**Cheeseman et al., 2013**; **Wordeman et al., 2016**), we compared kinetochore parameters after acute removal of PRC1 with those obtained from cells after long-term depletion of PRC1 by siRNA. Strikingly, in contrast to acute removal, the long-term depletion did not cause kinetochore misalignment or misorientation (**Figure 2D,H**; **Figure 2—figure supplement 1H,I**; **Table 1**). The two methods decreased the inter-kinetochore distance to a similar extent (**Figure 2G**; **Figure 2—figure supplement 1J**; **Table 1**), even though unlike acute removal, long-term removal reduced the fraction of cells that entered anaphase (35 ± 8%, N = 37, and 79 ± 6%, N = 37, for PRC1 siRNA treated and untreated, respectively; p = 0.046, Pearson's Chi-squared test; **Figure 1—figure supplement 1C**). Thus, acute removal of PRC1 results in different effects in comparison with a long-term depletion.

## PRC1 removal during metaphase increases the frequency of lagging kinetochores in anaphase

To test to what extent the acute removal of PRC1 during metaphase affects chromosome segregation, we measured the frequency of lagging kinetochores. We found that lagging kinetochores occurred more frequently when opto-PRC1 was being removed than in control cells (**Figure 2I,J**; **Figure 2—video 2**). Similarly, long-term depletion of PRC1 by siRNA also increased the frequency of lagging kinetochores during early anaphase (**Figure 2—figure supplement 2A**; **Table 1**).

Opto cells that showed lagging kinetochores in anaphase had a slightly smaller inter-kinetochore distance before anaphase than opto cells without lagging kinetochores (**Figure 2—figure supplement 2B**), suggesting that a decrease in tension may be involved in the imperfect kinetochore segregation. The cells with lagging kinetochores did not have a larger average kinetochore misalignment in metaphase (**Figure 2—figure supplement 2C**), which indicates that misalignment and lagging kinetochores are not linked on the cell level, although they may be linked locally on individual kinetochores.

**Table 1.** Comparison of effects of acute optogenetic removal of PRC1 and long-term depletion by siRNA.

All values are given as mean ± s.e.m. The numbers in the brackets denote the number of measurements and cells, respectively; a single number is the number of cells. Symbols (arrows and equal signs) denote trend of change of parameters; equal sign means no change; two arrows mark stronger effect. Measurements include cells without and with SiR-tubulin. Signal intensities of microtubule-associated proteins were normalized to the mean value of the signal of corresponding control for each treatment. *BAC* denotes live-cell experiments on HeLa cells expressing a fluorescently tagged protein from a BAC, *immuno* denotes immunocytochemistry experiments. [#]Consistent with our previous studies (*Kajtez et al., 2016*; *Polak et al., 2017*).

| Parameter | Acute removal | | | | Long-term depletion | | | |
|---|---|---|---|---|---|---|---|---|
| | 0 min | 20 min | p-value | | Untreated | siRNA | p-value | |
| $d_{EQ}$ (µm) | 0.78 ± 0.04 (290, 17) | 0.94 ± 0.05 (240, 16) | 0.035 | ↑ | 0.69 ± 0.05 (107, 8) | 0.69 ± 0.03 (333, 17) | 0.64 | = |
| $d_{EQ}$ > 2 µm (%) | 4.1 ± 1.2 (290, 17) | 8.3 ± 1.8 (240, 16) | 0.043 | ↑ | 3.3 ± 1.0 (107, 8) | 0.9 ± 0.9 (333, 17) | 0.33 | = |
| $d_{KC}$ (µm) | 0.87 ± 0.01 (226, 18) | 0.79 ± 0.01 (186, 18) | $1*10^{-8}$ | ↓ | 0.85 ± 0.01 (75, 8)[#] | 0.780 ± 0.008 (202, 17)[#] | $4*10^{-6}$ | ↓ |
| $\alpha_{KC}$ (°) | 13.8 ± 0.7 (290, 17) | 18.4 ± 1.0 (240, 16) | 0.0013 | ↑ | 13.3 ± 1.0 (107, 8) | 10.3 ± 0.7 (333, 17) | $1*10^{-4}$ | ↓ |
| $\alpha_{KC}$ > 35° (%) | 6.2 ± 1.4 (290, 17) | 12.1 ± 2.1 (240, 16) | 0.018 | ↑ | 4.7 ± 2.0 (107, 8) | 3.6 ± 1.0 (333, 17) | 0.83 | = |
| Lagging kinetochores (%) | 23 ± 7 (39) ctrl | 46 ± 9 (28) opto | 0.044 | ↑ | 5 ± 4 (37) | 40 ± 11 (20) | 0.0036 | ↑ |
| EB3 comets (min$^{-1}$) | 1.8 ± 0.1 (47, 5) ctrl | 2.0 ± 0.3 (26, 3) opto | 0.59 | = | 1.91 ± 0.09 (144, 6) | 1.5 ± 0.1 (54, 4) | 0.01 | ↓ |
| EB3 comets velocity (µm/min) | 19 ± 1 (17, 9) ctrl | 19.6 ± 0.8 (23, 9) opto | 0.73 | = | 16.0 ± 0.6 (16, 6) | 14.7 ± 0.8 (14, 4) | 0.19 | = |
| Half-overlap length (µm) | 1.8 ± 0.2 (17, 9) ctrl | 2.6 ± 0.2 (23, 9) opto | 0.009 | ↑ | 2.0 ± 0.2 (16, 6) | 1.8 ± 0.2 (14, 4) | 0.45 | = |
| N (MTs) in the bridging fiber | 14 ± 2 (22, 13) | 5.6 ± 0.9 (19, 10) | 0.0038 | ↓ | 14 ± 2 (18, 15) | 7.5 ± 1.1 (26, 18) | 0.0028 | ↓ |
| Curvature (µm$^{-1}$) | 0.134 ± 0.004 (40, 10) | 0.081 ± 0.005 (39, 10) | $6*10^{-11}$ | ↓↓ | 0.137 ± 0.006 (20, 5) | 0.108 ± 0.004 (52, 13) | 0.0014 | ↓ |
| θ (°) | 142 ± 3 (20, 10) | 125 ± 3 (20, 10) | $2*10^{-4}$ | ↓ | 145 ± 3 (10, 5) | 137 ± 2 (26, 13) | 0.051 | = |
| Kif4A intensity in the bridging fiber | 1 ± 0.1 (54, 11) | 0.24 ± 0.05 (32, 6) | $1*10^{-4}$ | ↓↓ | immuno 1 ± 0.03 (72, 17) | 0.48 ± 0.02 (58, 15) | $1*10^{-4}$ | ↓ |
| | | | | | BAC 1 ± 0.1 (43, 12) | 0.35 ± 0.04 (43, 11) | $1*10^{-4}$ | ↓ |
| Kif18A intensity in the bridging fiber | 1 ± 0.1 (57, 15) | 0.57 ± 0.09 (28, 11) | 0.01 | ↓ | immuno 1 ± 0.07 (121, 33) | 0.62 ± 0.05 (89, 40) | $1*10^{-4}$ | ↓ |
| | | | | | BAC 1 ± 0.1 (38, 14) | 0.53 ± 0.11 (16, 6) | 0.009 | ↓ |
| MKLP1 intensity in the bridging fiber | 1 ± 0.19 (25, 6) | 0.13 ± 0.05 (20, 6) | 0.008 | ↓ | 1 ± 0.22 (29, 8) | 0.17 ± 0.10 (20, 6) | 0.008 | ↓ |

As perturbation of the PRC1-CLASP1 interaction and the consequent absence of CLASP1 from the spindle midzone results in lagging chromosomes (*Liu et al., 2009*), we inspected the localization of CLASP1 and found that it did not accumulate between segregating chromosomes in opto HeLa cells stably expressing EGFP-CLASP1 (*Figure 2—figure supplement 2D*). Thus, the observed higher occurrence of lagging kinetochores could be attributed to changes in tension during metaphase, perturbed recruitment of CLASP1 to the spindle midzone by PRC1 during early anaphase, or a combination of both effects.

## Acute PRC1 removal leads to longer antiparallel overlap zones within the bridging fibers

Factors that could contribute to the altered chromosome alignment and occurrence of lagging kinetochores upon opto-PRC1 removal are changes related to (1) microtubules in the bridging fibers, (2) polar ejection forces, and/or (3) proteins that modulate the dynamics of k-fiber plus-ends (*Figure 2K*). Because PRC1 crosslinks microtubules within bridging fibers (*Kajtez et al., 2016*; *Polak et al., 2017*), we first tested the effects of PRC1 removal on those microtubules.

An important aspect of the bridging fiber that may affect chromosome alignment is the dynamics of microtubules that make up these fibers. To explore their dynamics, we developed an assay to track the growing plus ends of individual microtubules in the bridging fiber by using cells expressing the plus end marker EB3 (*Stepanova et al., 2003*) tagged with GFP (*Figure 3*). We followed single EB3 spots in the spindle and identified the ones belonging to a bridging fiber as the spots that move toward a kinetochore, cross the region between this kinetochore and its sister, and move beyond it toward the other spindle pole (*Figure 3A–C*; *Figure 3—video 1*). We found 1.8 ± 0.2 EB3 spots per minute per bridging fiber in control cells, showing that bridging fibers are dynamically remodeled during metaphase (*Figure 3D*). This number was similar after opto-PRC1 removal, 2.0 ± 0.3 spots/min (p = 0.59; t-test, *Figure 3D*), suggesting that the number of dynamic microtubules in the bridge is largely unaffected by acute PRC1 removal. However, in PRC1 siRNA-treated cells fewer EB3 spots were observed to move to opposite spindle half, 1.5 ± 0.1 EB3 spots/min, compared to untreated cells, where 1.91 ± 0.09 spots/min were observed (p = 0.01; t-test, *Figure 3—figure supplement 1A*), indicating that long-term PRC1 depletion slightly decreases the number of dynamic microtubules in the bridge.

To assess the changes in the dynamics of bridging microtubules, we followed EB3 spots in the bridge from the time when they can be distinguished from neighboring spots near the pole until they disappear in the opposite spindle half, which we interpret as the moment when the microtubule stops growing (*Maurer et al., 2012*). Interestingly, EB3 tracks were longer after opto-PRC1 removal than in control cells (*Figure 3B,C*; *Figure 3—video 1*), but the velocities of the EB3 spots were not affected by opto-PRC1 removal or long-term depletion when compared to untreated cells (*Figure 3E*; *Figure 3—figure supplement 1B*). In agreement with these results, kymographs of the central region of the spindle show that EB3 spots reach deeper into the opposite half of the spindle after opto-PRC1 removal (*Figure 3F*; *Figure 3—figure supplement 1C*). Thus, acute removal of PRC1 results in longer bridging microtubules, which is not a consequence of altered microtubule growth rate, but most likely due to a reduced microtubule catastrophe rate.

EB3 tracks allowed us to estimate the length of the overlap zone of antiparallel microtubules in the bridging fiber, which currently cannot be measured after PRC1 removal because PRC1 itself is the only available marker of antiparallel overlaps in the spindle. We define the overlap half-length as the distance beyond the equatorial plane that an EB3 spot covers while moving within the bridging fiber from one spindle half into the other (*Figure 3G*, left), where this microtubule can form antiparallel overlaps with the oppositely oriented microtubules extending from the other pole. In control cell, the overlap half-length was 1.8 ± 0.2 μm (n = 17 spots in N = 9 cells; *Figure 3G*, right), which corresponds well to the overlap half-length measured as the half-length of PRC1 streaks in this cell line, 0.5x(3.8 ± 0.4) μm (N = 9 cells; *Figure 3H*), validating our method for overlap length measurement based on EB3 tracks. After opto-PRC1 removal, the overlap half-length increased to 2.6 ± 0.2 μm (n = 23 spots in N = 9 cells; *Figure 3G*). In contrast to opto cells, treatment with PRC1 siRNA did not result in a change of overlap length with respect to untreated cells (*Figure 3—figure supplement 1D–F*). Thus, the antiparallel overlaps became on average 40% longer after acute, but unchanged after long-time PRC1 removal. Interestingly, the overlaps were especially long in the inner part of the spindle, whereas those on the spindle periphery were similar to overlaps in control cells (*Figure 3I*). This spatial difference is correlated with our findings that PRC1 is removed faster from the inner bundles (see *Figure 1B*) and that displaced and misoriented kinetochores are found more often in the inner than in the outer spindle region (see *Figure 2D,E,H*), suggesting a mechanistic link between the overlap length and kinetochore positioning.

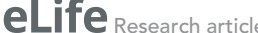

**Figure 3.** Acute removal of PRC1 elongates antiparallel overlaps within the bridging fibers. (A) Schematic of the trajectory of an EB3-marked plus end (white circles) within the bridging fiber, defined as a spot passing the sister kinetochore region (magenta). (B) Spindles (left) in U2OS cells with stable expression of 2xGFP-EB3 (gray) and mCherry-CENP-A (magenta), depleted for endogenous PRC1, with transient expression of opto-PRC1 and iLID-CAAX (opto; top) and opto-PRC1 only (control; bottom). Montage of the boxed region over time is shown as merged channels (middle) and GFP (right; the tracked spot is encircled). The cell was exposed to the blue light for 15 min in total. In the first ten minutes PRC1 was removed from the spindle applying the usual imaging protocol for opto cells. During the last five minutes of exposure to the blue light, faster image acquisition was used in order to see and track the EB3 dynamics (see Materials and methods). Note that in opto cell (top) PRC1 is removed from the spindle. Images are single z-planes smoothed with 0.5-pixel-sigma Gaussian blur. (C) Trajectories of tracked EB3 spots (connected dots) in opto (top) and control (bottom) cells. Black dot; start of trajectory. Single dots on the sides; spindle poles. (D) Number of EB3 spots per minute within the bridging fiber in opto (black) and control (gray) cells. (E) EB3 spot velocity within the bridging fiber in opto (black) and control (gray) cells. (F) Kymographs of opto (top) and control (bottom) cells after 10 min of imaging protocol required for removal of PRC1, merge (top) and GFP (bottom). Cyan arrowheads mark the beginning and end of an individual EB3 spot trajectory. Note the difference in the position of indicated track ends with respect to the equatorial plane. (G) Half-overlap length (left) defined as the distance (double arrow) between the end-point of the EB3 spot trajectory and the equatorial plane (dashed line). Black arrowhead; start of trajectory. Half-overlap length (right) in opto (black) and control (gray) cells measured as in scheme (left) and retrieved from tracks shown in B (right). (H) Metaphase plate width in opto (black) and control (gray) cells measured from kymographs as the largest distance between kinetochore pairs positioned on the opposite sides of the spindle equator in the first two minutes after 10 min of imaging protocol required for removal of PRC1. As kinetochores remain within the PRC1-labeled region in control cells, the metaphase plate width in these cells was measured as the PRC1 streak length. (I) Half-overlap length in opto cells for inner ($d_{AX} \leq 2$ μm) and outer ($d_{AX} > 2$ μm) overlaps. Filled cyan rectangles indicate exposure to the blue light. Numbers in brackets denote measurements and cells; single numbers denote cells. Error bars; s.e.m. Scale bars; 2 μm. Statistical analysis; t-test. p-value legend as in *Figure 2*.

The online version of this article includes the following video and figure supplement(s) for figure 3:

*Figure 3 continued on next page*

*Figure 3 continued*

**Figure supplement 1.** Long-term PRC1 depletion shows no effect on the length of antiparallel overlaps.
**Figure 3—video 1.** U2OS cell stably expressing 2x-GFP-EB3 (gray) and mCherry-CENP-A (magenta), with transient expression of opto-PRC1 (magenta; control, top), and opto-PRC1 (magenta), and iLID-CAAX (opto, bottom).
https://elifesciences.org/articles/61170#fig3video1

## Removal of PRC1 reduces the number of microtubules in the bridging fibers

To test to what extent PRC1 removal in metaphase affects the number of microtubules in the bridging fibers (*Figure 4A*), we visualized microtubules by using SiR-tubulin, a far-red tubulin dye excited by red light (*Lukinavičius et al., 2014*), which allowed us to observe microtubules both when the blue light is turned on and switched off. Intensity profiles across the spindle midzone revealed that SiR-tubulin intensity maxima were lower upon PRC1 removal and increased after its return (*Figure 4—figure supplement 1A*; *Figure 4—video 1*). Measurements of SiR-tubulin signal intensity between and lateral from sister kinetochores showed that upon PRC1 removal the tubulin signal was reduced specifically in the bridging fibers (*Figure 4A–C*; *Figure 4—video 1*). As an alternative to SiR-tubulin, which is a taxol-based dye that may affect microtubule dynamics (*Lukinavičius et al., 2014*), we tested YFP-tubulin, but the excitation laser for YFP also activated the optogenetic system (*Wang and Hahn, 2016*; *Figure 4—figure supplement 1B*).

Finally, we used tubulin-GFP to determine tubulin signal intensities of the bridging fibers and k-fibers upon acute removal of PRC1 (*Figure 4A,D–H*; *Figure 4—figure supplement 1C–F*; see Materials and methods). Upon exposure to the blue light, tubulin signal intensity in the bridging fibers decreased ~2.5-fold, which corresponds to $5.6 \pm 0.9$ microtubules given that the average number of microtubules in the bridging fiber is 14 (*Kajtez et al., 2016*). Together with the finding that the number of growing microtubules in the bridging fiber is similar with and without PRC1 (*Figure 3D*), this result implies that in the presence of PRC1 the bridge contains more microtubules with a smaller fraction of them being dynamic than without PRC1, and that PRC1 removal leads mainly to disassembly of non-dynamic microtubules. Upon PRC1 return the intensity and thus the number of microtubules remained low (*Figure 4D–G*; *Figure 4—figure supplement 1D*), possibly as a consequence of the perturbed spindle architecture in the absence of PRC1, which may be able to recover after a longer time period. Importantly, the intensity of k-fiber was unaltered (*Figure 4H*, see Materials and methods), suggesting that the k-fibers were not affected by PRC1 removal. Thus, acute removal of PRC1 and its return change the number of microtubules specifically in the bridging fibers.

Long-term depletion of PRC1 by siRNA in HeLa cells with stable expression of tubulin-GFP and transient expression of mRFP-CENP-B resulted in a similar reduction in bridging fiber tubulin intensity as after acute PRC1 removal (*Figure 4—figure supplement 1G,H*). Long-term depletion resulted in a decreased number of microtubules in the bridge, $7.5 \pm 1.1$, which was similar to the microtubule number after acute removal ($p = 0.21$, t-test; *Figure 4—figure supplement 1I*, *Table 1*). As in acute removal, the intensity of k-fibers did not change after long-term depletion ($p = 0.49$, t-test).

Given that the bridging fiber is under compression (*Kajtez et al., 2016*), reduction of number of microtubules in the bridging fiber is expected to reduce this compression and thus to straighten the k-fibers (*Figure 4—figure supplement 1J*). Tracking of the pole-to-pole contour of the outermost k-fibers revealed that PRC1 removal indeed straightened the k-fibers and thus made the spindle diamond-shaped (*Figure 4—figure supplement 1K–M*). Similar to acute removal, long-term depletion of PRC1 caused straightening of outermost k-fibers, although to a smaller extent, whereas spindle length and width were unchanged after both treatments (*Figure 4—figure supplement 1L,N,O*; *Table 1*). This result supports that compression in the bridging fibers enables the spindle to obtain a curved shape.

Since bridging fibers are, nevertheless, present upon PRC1 removal, we asked how the residual microtubules are bundled together. Eg5/kinesin-5, which localizes in the bridging fibers (*Kajtez et al., 2016*; *Mann and Wadsworth, 2018*), was still detectable in these fibers after PRC1



**Figure 4.** Optogenetic removal of PRC1 reduces bridging fibers. (**A**) Schematic of PRC1 (magenta) removal from bridging fibers and positions where bridging fiber and bridging and k-fiber tubulin (green) intensities were measured (dashed lines). (**B**) Spindle in a U2OS cell with stable expression of CENP-A-GFP (cyan), depleted for endogenous PRC1, with transient expression of opto-PRC1 (magenta) and iLID-CAAX, and stained with SiR-tubulin (green), before exposure to the blue light (0 min, Dark), at the end of continuous exposure to the blue light (20 min, Light), and 10 min after cessation of exposure to the blue light (30 min, Dark). Enlargements of the boxed region (first: merge, second: SiR-tubulin, third: opto-PRC1, fourth: CENP-A-GFP) are shown. Yellow lines represent the positions where bridging fiber and bridging and k-fiber intensities were measured. Images are a single z-plane smoothed with 0.5-pixel-sigma Gaussian blur. (**C**) Background-corrected SiR-tubulin intensity profiles of the bridging fiber (left) and bridging and k-fiber (right) for cell shown in B (0 min, black; 20 min, cyan; 30 min, gray). (**D**) Spindle in a HeLa cell with stable expression of tubulin-GFP (green), depleted for endogenous PRC1, with transient expression of opto-PRC1 (magenta) and iLID-CAAX, and stained with SiR-DNA (not shown), before exposure to the blue light (0 min, Dark, left), at the end of continuous exposure to the blue light (20 min, Light, middle) and 10 min after cessation of exposure to the blue light (30 min, Dark, right). Enlargements of the boxed region (middle: merge, bottom: tubulin-GFP) are shown. Note that at 20 min opto-PRC1 is absent from the spindle. (**E**) Graphs showing tubulin-GFP intensity profiles of the bridging fiber (left) and bridging and k-fiber (0 min, black; 20 min, cyan; 30 min, gray) for cell shown in D. Horizontal line marks the background signal, vertical dashed lines delimit the area (shaded) where signal was measured. (**F**) Enlargements of spindles in HeLa cells with stable expression of tubulin-GFP (green), depleted for endogenous PRC1, with transient expression of opto-PRC1 (magenta) and iLID-CAAX, and stained with SiR-DNA (not shown), before exposure to the blue light (0 min, Dark, top; first row: merge, second row: tubulin-GFP) and at the end of continuous exposure to the blue light (20 min, Light, bottom; first row: merge, second

*Figure 4 continued on next page*

*Figure 4 continued*

row: tubulin-GFP). Note that at 20 min opto-PRC1 is absent from the spindle. Images do not belong to the same cell. Asterisks mark the position of kinetochores. All images are single z-plane smoothed with 0.5-pixel-Gaussian blur. Scale bar; 2 µm. (G) Number of microtubules in the bridging fiber in opto HeLa cells (that is, where opto-PRC1 was removed; black) and control (gray) in same time-points as in D. The bridging fiber intensity in control cells before exposure to the blue light is set to correspond to 14 microtubules (see *Figure 4—figure supplement 1D*). (H) Tubulin-GFP signal of k-fibers in opto (black) and control (gray) HeLa cells at time-points as in D. Cyan rectangles inside graphs indicate exposure to the blue light. Numbers in brackets; number of measurements and cells, respectively. Error bars; s.e.m. Scale bars; 2 µm. Statistical analysis; one-way ANOVA followed by Tukey HSD post hoc test, t-test (G, H). p-value legend as in *Figure 2*.

The online version of this article includes the following video and figure supplement(s) for figure 4:

**Figure supplement 1.** Reduction of bridging fibers straightens the spindle contour.

**Figure 4—video 1.** U2OS cell stably expressing CENP-A-GFP (not shown), with transient expression of opto-PRC1 (magenta) and iLID-CAAX, and microtubules stained with SiR-tubulin (green).

https://elifesciences.org/articles/61170#fig4video1

siRNA (*Figure 4—figure supplement 1P*). Thus, we propose that microtubule crosslinkers such as Eg5 crosslink the remaining microtubules in the bridge after acute PRC1 removal.

## Kif4A, Kif18A, and MKLP1 localize in the bridge during metaphase in a PRC1-dependent manner

To investigate the mechanism of bridging microtubule regulation via PRC1, we analyzed the localization of major proteins that regulate spindle microtubule dynamics and/or are binding partners of PRC1: Kif4A, Kif18A, and MKLP1 (*Bieling et al., 2010b*; *Bringmann et al., 2004*; *Gruneberg et al., 2006*; *Kurasawa et al., 2004*; *Mayr et al., 2007*; *Stumpff et al., 2008*; *Stumpff et al., 2012*) before and after PRC1 removal in metaphase (*Figure 5*; *Figure 5—figure supplement 1*; *Figure 5—figure supplement 2* and *Table 1*).

We first looked at the localization of these proteins in the bridging fiber. Surprisingly, we found that Kif4A and Kif18A localize in the bridge during metaphase, visible as thin lines across or next to the location of sister kinetochores where PRC1-labeled bundles are found (*Polak et al., 2017*; *Figure 5A*; *Figure 5—figure supplement 1A*). Similar localization was observed for MKLP1, spanning the region between two sister k-fibers (*Figure 5A*). In vertically oriented spindles, whose long axis was roughly perpendicular to the imaging plane, Kif4A and Kif18A colocalized with opto-PRC1, now visible as spots in the cross-section of the spindle (*Figure 5—figure supplement 1B*).

To explore whether PRC1 removal affects the localization of Kif4A, Kif18A, and MKLP1, we analyzed their intensities within bridging fibers. Measurements of Kif4A intensity upon acute PRC1 removal and long term-depletion, in regions lateral from chromosomes which correspond to peripheral parts of bridging fibers, showed that its intensity in the bridging fibers decreased (*Figure 5A–C*; *Figure 5—figure supplement 1A*). Additional analysis of Kif4A intensity in HeLa cells after long-term PRC1 removal corroborated this result (*Figure 5—figure supplement 1C*). Interestingly, there was a larger reduction in Kif4A intensity after acute PRC1 removal, where it decreased by 76 ± 5%, in comparison to long-term depletion where it decreased by 52 ± 3% (p = $1\times10^{-4}$; *Figure 5C*; *Table 1*). Kif18A intensity in the bridging fibers also decreased upon acute PRC1 removal and long-term depletion (*Figure 5A–C*; *Figure 5—figure supplement 1A,C*). Yet, in contrast to Kif4A, Kif18A intensities in the bridging fiber decreased to a similar extent, that is by 43 ± 11% and 38 ± 6%, after acute and long-term PRC1 depletion, respectively (p = 0.68; *Figure 5C*; *Table 1*). MKLP1 intensities were also similarly reduced after both approaches, 87 ± 5% after acute PRC1 removal and 83 ± 11% after long-term depletion (p = 0.68; *Figure 5D*; *Figure 5—figure supplement 1C*).

Among the tested proteins, only MKLP1 was exclusively localized in the bridging fibers, co-localizing with PRC1, both before PRC1 removal and after its return (*Figure 5A,D*; *Figure 5—figure supplement 1C–G*). Even though MKLP1 was removed to a large extent from the spindle by acute PRC1 removal, it was not detected at the cell membrane together with PRC1. It may be that MKLP1 binds rather weakly to PRC1 in metaphase and/or that the absence of PRC1 decreases its affinity for microtubules. In addition, the ability of MKLP1 to bind along scaffold of antiparallel overlaps could depend on the role of PRC1 in dictating 35-nm-inter-microtubule spacing, proposed to be important to enable localization of specific proteins within these structures (*Kellogg et al., 2016*; *Subramanian et al., 2010*).

**Figure 5.** Localization of Kif4A, Kif18A, and MKLP1 after acute removal and long-term depletion of PRC1. (**A**) Spindle (first block, left) in a U2OS cell with stable expression of CENP-A-GFP (not shown), immunostained for endogenous Kif4A (AF-594, green), PRC1 (AF-647, magenta) and stained with DAPI (cyan). Enlargements of the boxed region are shown (right). White arrowhead points to Kif4A outside chromosomes, at the position where PRC1-AF-647 is found, which corresponds to the bridging fiber. Spindle (middle block, top) in a U2OS cell with stable expression of CENP-A-GFP (cyan),

*Figure 5 continued on next page*

*Figure 5 continued*

immunostained for endogenous Kif18A (AF-594, green) and PRC1 (AF-647, magenta). Enlargements of the boxed region are shown (bottom). White arrowhead points to Kif18A in the bridging fiber where PRC1-AF-647 is found. Spindle (right block, top) of HeLa cell stably expressing MKLP1-GFP (green) and stained for SiR-tubulin (magenta). Enlargements of the boxed region are shown (bottom). White arrowhead points to the MKLP1 in the bridging fiber. Schemes (right) show Kif4A and Kif18A localization on chromosome arms and plus ends of k-fibers, respectively, and Kif4A, Kif18A, and MKLP1 in the bridging fiber. (B) Time-lapse images (left block) of unlabeled U2OS cell with transient expression of opto-PRC1 (magenta), iLID-CAAX and GFP-Kif4A (green), and stained with SiR-DNA (cyan) before (Dark, top) and after 12 min of the exposure to the blue light (Light, middle). Enlargements of the boxed regions are shown (bottom rows). Before opto-PRC1 removal (Dark) Kif4A is also found outside chromosomes, at the position of opto-PRC1 labeled bundles (white arrowheads). Note that after 12 min of opto-PRC1 removal (Light), Kif4A signal is found only at the positions of the chromosomes. The intensities in the enlargements are adjusted differently than those of the whole spindle to better point out localization of proteins. Time-lapse images (right block) of unlabeled U2OS cell with transient expression of opto-PRC1 (magenta), iLID-CAAX and EGFP-Kif18A (green) before (Dark, top) and after 12 min of the exposure to the blue light (Light, middle). Enlargements of the boxed regions are shown (bottom). Before opto-PRC1 removal (Dark, bottom left) Kif18A is found in the bridging fiber (white arrowhead). In the enlargement of the boxed region after 12 min of opto-PRC1 removal (Light, bottom right), the signal of the EGFP-Kif18A on the bridging fiber is weaker. (C) Normalized bridging fiber Kif4A (left) and Kif18A (right) intensity measured before and after acute and long-term PRC1 removal in cells as in B and A, respectively. Dark; black, Light; cyan, untreated; magenta, PRC1 siRNA; green. (D) Timelapse of the spindle in HeLa BAC cell (left) stably expressing MKLP1-GFP (gray) with transient expression of opto-PRC1 (not shown) and iLID-CAAX, and stained with SiR-DNA (not shown) before (0 min, Dark), at the end of continuous exposure (20 min, Light) and 10 min after cessation of exposure to the blue light (30 min, Dark). Note that opto-PRC1 is not shown in order to point out localization of MKLP1-GFP. Image is a maximum projection of three z-planes. Graph (right) shows normalized bridging fiber MKLP1 intensity measured before and after acute and long-term PRC1 removal in cells as in *Figure 5—figure supplement 1C*. (E) Spindles from HeLa cells stably expressing PRC1-GFP (gray) in untreated (left), Kif4A siRNA (middle), and Kif18A siRNA (right) -treated cell. (F) Ratios of overlap and spindle lengths for untreated (black), Kif4A siRNA (magenta), and Kif18A siRNA (green) -treated cells. Gray scattered points show individual measurements. (G) Difference in PRC1-labeled overlap and spindle length for Kif4A siRNA (magenta) and Kif18A siRNA (green) treatment when compared to untreated cells. Numbers in brackets; number of measurements and cells, respectively. Statistical analysis: t-test (C, G); one-way ANOVA (F). All images are smoothed with 0.5-pixel-sigma Gaussian blur, and one z-plane is shown unless stated otherwise. Scale bars: 2 μm.

The online version of this article includes the following figure supplement(s) for figure 5:

**Figure supplement 1.** Kif4A, Kif18A, and MKLP1 show PRC1-dependent localization in the bridging fibers.

**Figure supplement 2.** PRC1 removal shows no effect on polar ejection forces and k-fiber plus end dynamics.

As Kif4A and Kif18A are known to regulate microtubule length (*Mayr et al., 2007*; *Varga et al., 2006*; *Wandke et al., 2012*), we set out to explore their roles in overlap length regulation. We hypothesized that these kinesins may regulate the length of bridging microtubules and hence their antiparallel overlaps during metaphase. Indeed, removal of Kif4A or Kif18A by siRNA resulted in longer PRC1-labeled overlaps and longer spindles (*Figure 5E*), with a small but not significant increase in the ratio of overlap length to spindle length (*Figure 5F*). However, the increase in the overlap and spindle length was significantly different after the two treatments (*Figure 5G*). Kif4A siRNA treatment increased overlap length for 0.9 ± 0.2 μm compared with control, which was similar to spindle length increase of 1.2 ± 0.3 μm (p = 0.48, t-test; *Figure 5G*). In contrast, Kif18A siRNA treatment increased overlap length for 1.8 ± 0.3 μm, whereas the spindle length increased to a larger extent, for 3.7 ± 0.6 μm (p = 0.004, t-test; *Figure 5G*). These results suggest that both Kif4A and Kif18A regulate the length of bridging microtubules and their overlap, but Kif18A has a greater effect on overall spindle microtubules and spindle length than Kif4A.

## Kif4A persists on chromosome arms and Kif18A, CLASP1, and CENP-E on kinetochore fiber tips upon PRC1 removal

As an alternative to the changes in the bridging fiber, disruption of polar ejection forces may be the cause of kinetochore misalignment upon PRC1 removal (*Figure 2K*), in particular because these forces are modulated by Kif4A (*Stumpff et al., 2012*). The most prominent Kif4A localization in metaphase is on chromosomes (*Gruneberg et al., 2006*; *Kurasawa et al., 2004*; *Zhu and Jiang, 2005*; *Figure 5A,B*; *Figure 5—figure supplement 1A–C*; *Figure 5—figure supplement 2A–C*). Neither acute nor long-term PRC1 removal had an extensive effect on the Kif4A signal on chromosome arms, although there was a slight decrease in Kif4A signal after PRC1 siRNA treatment (*Figure 5B*; *Figure 5—figure supplement 1A,C*; *Figure 5—figure supplement 2C,D*; *Table 1*). In early anaphase, the amount of Kif4A on segregated chromosomes was similar in opto and control cells (*Figure 5—figure supplement 2E*), indicating that increased occurrence of lagging kinetochores was

neither due to perturbed polar ejection forces in that phase nor defects in chromosome architecture/condensation.

Finally, kinetochore misalignment and lagging kinetochores upon PRC1 removal may be due to disrupted localization of proteins that regulate microtubule dynamics at the plus-end of the k-fiber (*Figure 2K*). To investigate this possibility, we analyzed the k-fiber localization of the key proteins with this function, Kif18A, CLASP1, and CENP-E (*Al-Bassam et al., 2010*; *Maffini et al., 2009*; *Maiato et al., 2003*; *Mayr et al., 2007*; *Stumpff et al., 2008*; *Stumpff et al., 2012*; *Yu et al., 2016*). These proteins were localized at the plus-ends of k-fibers before and after acute and long-term removal of PRC1 (*Figure 5A,B*; *Figure 5—figure supplement 1A–C*; *Figure 5—figure supplement 2F–I*). This indicates that kinetochore misalignment upon PRC1 removal is not caused by simultaneous removal of proteins that regulate microtubule dynamics at the k-fiber plus ends.

## Discussion

### An optogenetic system for acute, selective, and reversible removal of spindle proteins

We developed an optogenetic approach that offers acute light-controlled removal of proteins from a normally formed spindle at a precise phase of mitosis. The main advantages of this approach over chemically-induced protein translocation (*Cheeseman et al., 2013*; *Haruki et al., 2008*; *Robinson et al., 2010*; *Wordeman et al., 2016*) are its reversibility, allowing the protein to return to its initial location within about a minute, and applicability to individual cells. Unlike previous optogenetic approaches (*Fielmich et al., 2018*; *Okumura et al., 2018*; *van Haren et al., 2018*; *Yang et al., 2013*; *Zhang et al., 2017*), this method allows for global loss-of-function of full-length spindle proteins, relying on simple protein tagging rather than domain splitting, with no need of chromophore addition. Moreover, this method may be implemented with other optical perturbations (*Milas et al., 2018*) and used as '*in vivo* pull-down' for probing protein-protein interactions in different phases of the cell cycle. However, this approach depends on high turnover of the protein in comparison with the time scale of interest.

Acute PRC1 removal from the spindle by optogenetics and long-term PRC1 depletion by siRNA led to partially different phenotypes. These differences are hard to explain by different levels of PRC1 on the spindle as both methods decreased PRC1 by ~90%. It is also unlikely that the differences are caused by the interaction of the membrane-translocated PRC1 with astral microtubules because of the uniform PRC1 signal on the membrane, fast return to the spindle, and no change in spindle positioning. Therefore, the generally weaker effects of siRNA in comparison with the acute optogenetic removal are most likely due to compensatory mechanisms acting during long-term depletion.

### A model for chromosome alignment by overlap-length-dependent forces within the bridging fibers

By overcoming temporal limitations of siRNA, our work reveals an unexpected role of PRC1 and bridging fibers in the regulation of chromosome alignment on the metaphase spindle via overlap length-dependent forces (*Figure 6A*). We propose that the interactions between k-fibers and bridging fibers regulate the movement of bi-oriented chromosomes along the pole-to-pole axis. If a kinetochore pair is displaced toward one spindle pole, more motors and/or crosslinkers are expected to accumulate in the overlap facing the opposite pole because this overlap is longer, and pull the kinetochores back to the center (*Figure 6B*). As the efficiency of this type of centering depends on the relative asymmetry in the overlap length on either side, shorter overlap length leads to more precise centering of the kinetochores (*Figure 6A,B*).

Our finding that acute PRC1 removal leads to chromosome misalignment and elongation of overlap zones supports this model. Moreover, elongation of overlaps correlates with chromosome misalignment within spindles, as elongated overlaps and misaligned chromosomes are mostly found in the central part of the spindle rather than on the periphery. In contrast to the acute PRC1 removal, long-term PRC1 depletion by siRNA leads neither to overlap elongation nor chromosome misalignment, providing further support to our model. These differences between the acute and long-term

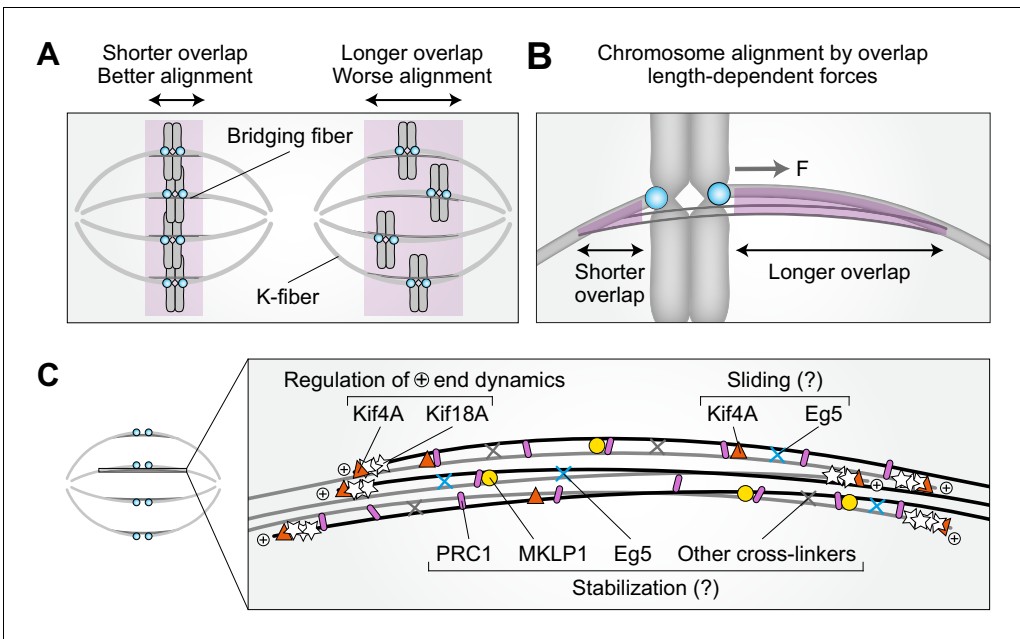

**Figure 6.** Model for chromosome alignment by overlap length-dependent forces within the bridging fiber. (**A**) We propose that the interactions between k-fibers and bridging fibers regulate the movement of bi-oriented chromosomes by forces that depend on the length of the antiparallel overlaps (purple). Shorter overlaps lead to more precise alignment of kinetochores (cyan) than longer ones. (**B**) If a kinetochore pair is displaced away from the equatorial plane toward one pole, the overlap between the k-fiber and the bridging fiber (purple) is shorter on this side and longer on the opposite side. More motors and/or crosslinkers accumulate in the longer overlap, pulling the kinetochore back to the center (*F*, pulling force). The efficiency of centering depends on the relative asymmetry in the overlap length on either side. This asymmetry is larger if the overlap is short, which explains why short overlaps lead to better alignment than long ones (see A). (**C**) The overlap length is regulated by Kif4A and Kif18A at the plus ends of bridging microtubules. Kif4A and Eg5 within the bridging fiber possibly slide the microtubules apart, whereas PRC1 stabilizes the overlaps probably together with MKLP1, Eg5, and other crosslinkers.

depletion indicate the existence of compensatory mechanisms that regulate overlap length and hence maintain chromosome alignment during long-term PRC1 depletion.

How is the length of the bridging microtubules and their overlaps controlled? Interestingly, we found that Kif4A and Kif18A localize in the bridging fibers in metaphase and their intensities decreased following partial disassembly of the bridging fiber by optogenetic or siRNA-mediated PRC1 removal. Both Kif4A and Kif18A suppress microtubule dynamics (*Bieling et al., 2010b*; *Bringmann et al., 2004*; *Stumpff et al., 2008*; *Stumpff et al., 2012*), and our experiments showed that depletion of either of them by siRNA leads to longer PRC1-labeled overlaps. These elongated PRC1-labeled overlaps provide another line of evidence in support of our model for chromosome alignment by overlap length-dependent forces, as Kif4A and Kif18A siRNA-treated cells exhibit chromosome misalignment (*Stumpff et al., 2008*; *Wandke et al., 2012*).

Even though it was reported that Kif4A does not directly interact with PRC1 before late mitosis (*Gruneberg et al., 2006*; *Kurasawa et al., 2004*; *Zhu and Jiang, 2005*), our experiments suggest that there is a small pool of Kif4A that does so and binds to the antiparallel overlaps of bridging microtubules in metaphase. Acute PRC1 removal likely results in the removal of this pool of Kif4A from the bridge. Our result that Kif4A was reduced by ~76%, which is similar to the reduction of PRC1 by ~85% within the same time interval (12 min) of acute removal supports this picture. This reduction of Kif4A in the bridge may lead to excessive microtubule polymerization and thus longer overlap zones in metaphase, similar to the long overlaps in anaphase after Kif4A depletion (*Hu et al., 2011*; *Kurasawa et al., 2004*; *Zhu and Jiang, 2005*). As the spindle length remained constant after acute PRC1 removal, we suggest that the overlap elongation is due to Kif4A acting specifically on bridging microtubules rather than overall spindle microtubules.

In contrast to Kif4A, Kif18A in the bridging fiber was reduced less than PRC1 within 12 min of acute PRC1 removal, ~43% compared with ~85%, respectively. This difference in the behavior of Kif18A and PRC1 is consistent with the fact that Kif18A is not known to be a binding partner of PRC1. Thus, we infer that the reduction of Kif18A in the bridge was a consequence of the fewer microtubules in the bridging fiber, and that the amount of Kif18A per microtubule remained largely unaffected by PRC1 removal. Finally, MKLP1, which is a binding partner of PRC1 (*Gruneberg et al., 2006*), was reduced to a similar extent as PRC1, ~87% and ~88% within 20 min of acute PRC1 removal, respectively. Based on these results, we conclude that the localization of MKLP1 and a small pool of Kif4A in the bridging fiber are directly PRC1-dependent, whereas the localization of Kif18A is not.

We propose that both Kif4A and Kif18A regulate the length of bridging microtubules and their overlaps under normal conditions (*Figure 6C*), as their depletion results in longer overlaps. Intriguingly, we found that Kif4A signal in the bridging fibers decreased to a larger extent after acute than after long-term PRC1 depletion, whereas Kif18A signal decreased to a similar extent by the two methods of PRC1 depletion. Moreover, Kif4A depletion by siRNA resulted in an increase of ~1 μm in overlap length, which was similar to the increase observed after acute PRC1 removal, whereas Kif18A depletion by siRNA led to a larger overlap increase. Thus, we suggest that the observed overlap elongation after acute PRC1 removal is a consequence of the strong reduction of Kif4A in the bridging fibers. In contrast, after long-term PRC1 depletion, we speculate that a larger fraction of Kif4A is present on the microtubules during overlap formation, contributing to the compensatory mechanism maintaining normal overlap length.

The proteins that we found in the bridging fiber, Kif4A, Kif18A, and MKLP1, may also slide apart and/or stabilize the bridging microtubules (*Figure 6C*). In addition to these kinesins, we observed Eg5 in the bridging fiber (*Kajtez et al., 2016*; *Mann and Wadsworth, 2018*). During early anaphase, Kif4A and Eg5 drive the sliding of antiparallel microtubules that elongates the spindle (*Vukušić et al., 2019*). Thus, these kinesins may have a similar role during metaphase. This possibility is in agreement with previous work showing that Kif4A depletion reduces microtubule poleward flux in metaphase (*Wandke et al., 2012*). Similarly, Kif18A in the bridging fiber may have microtubule sliding and crosslinking activities equivalent to those of the yeast kinesin-8 (*Su et al., 2013*). Furthermore, MKLP1 contributes to the stabilization of bridging microtubules in early anaphase (*Vukušić et al., 2017*), and we suggest that it performs a similar function in metaphase together with other motors and crosslinkers including Eg5 and PRC1. The roles of these and other proteins within bridging fibers in the regulation of microtubule dynamics and sliding will be an intriguing topic for future studies.

The localization of Kif4A on chromosome arms, where it is involved in polar ejection forces (*Bieling et al., 2010a*; *Brouhard and Hunt, 2005*) was preserved after acute PRC1 removal. Similarly, Kif18A, CLASP1, and CENP-E remained on plus-ends of k-fibers. However, as we cannot exclude potential subtle changes in the localization of these proteins or mislocalization of other proteins, the observed chromosome misalignment and overlap elongation after acute PRC1 removal could also be promoted by the changes of the dynamics of other microtubules within the spindle consequently affecting forces produced by k-fibers and polar ejection forces.

The changes upon acute PRC1 removal were not spatially uniform across the spindle. The most affected part was the inner part of the spindle, where the PRC1 signal disappeared faster and the bridging microtubules became longer than on the periphery of the spindle. The inner bridging fibers were more severely affected by PRC1 removal possibly because they are made up of fewer microtubules than the outer bridges. Severely misaligned kinetochores that moved more than 2 μm away from the equatorial plane and lagging kinetochores occurred also more often in the inner part of the spindle. This local effect is in line with weak mechanical coupling between neighboring k-fibers yet strong coupling between sister k-fibers (*Elting et al., 2017*; *Suresh et al., 2020*; *Vladimirou et al., 2013*), and indicative of a mechanistic link between the bridging fiber geometry and kinetochore alignment.

In conclusion, we propose that overlap length-dependent forces help to position the chromosomes at the equatorial plane of the spindle. The overlap length is regulated by the PRC1-dependent Kif4A, and by Kif18A within the bridging fiber. Kif4A, Eg5, and possibly other kinesins may slide bridging microtubules apart, whereas PRC1 together with the kinesins stabilizes the overlaps. Thus, in addition to the forces generated at k-fiber tips and polar ejection forces, proper

chromosome alignment requires forces generated within the bridging fiber, which are transferred to the k-fiber and rely on precise regulation of the overlap region.

# Materials and methods

## Key resources table

| Reagent type (species) or resource | Designation | Source or reference | Identifiers | Additional information |
|---|---|---|---|---|
| Cell line (*Homo sapiens*) | Unlabeled U2OS cells | Gift from Marin Barišić and Helder Maiato; used in our previous work (*Vukušić et al., 2017*) | | Human osteosarcoma cells |
| Cell line (*Homo sapiens*) | U2OS cells stably expressing CENP-A-GFP | Gift from Marin Barišić and Helder Maiato; used in our previous work (*Vukušić et al., 2017*) | | Human osteosarcoma cells |
| Cell line (*Homo sapiens*) | U2OS cells stably expressing CENP-A-GFP, mCherry-α-tubulin and PA-GFP-tubulin | Gift from Marin Barišić and Helder Maiato; used in our previous work (*Vukušić et al., 2017*) | | Human osteosarcoma cells |
| Cell line (*Homo sapiens*) | U2OS cell line stably expressing 2xGFP-EB3 and mCherry-CENP-A | Gift from Julie Welburn; used in our previous work (*Kajtez et al., 2016*) | | Human osteosarcoma cells |
| Cell line (*Homo sapiens*) | HeLa-Kyoto BAC MKLP1-GFP | Gift from Ina Poser and Tony Hyman | 4319 HeLa-ky KIF-23- hum MKLP1 T#280 | |
| Cell line (*Homo sapiens*) | HeLa-Kyoto BAC Kif4A-GFP; HeLa stably expressing Kif4A-GFP | Gift from Ina Poser and Tony Hyman | HeLa Kif4A-GFP 0005146 T#372 | |
| Cell line (*Homo sapiens*) | HeLa-Kyoto BAC Kif18A-GFP; HeLa stably expressing Kif18A-GFP | Gift from Ina Poser and Tony Hyman | HeLa Kif18A-GFP MCB 0003184 #197 | |
| Cell line (*Homo sapiens*) | HeLa-Kyoto BAC CENP-E-GFP; HeLa stably expressing CENP-E-GFP | Gift from Ina Poser and Tony Hyman | HeLa CENP-E-GFP T#363 | |
| Cell line (*Homo sapiens*) | HeLa-Kyoto BAC PRC1-GFP; HeLa stably expressing PRC1-GFP | Gift from Ina Poser and Tony Hyman | | |
| Cell line (*Homo sapiens*) | Unlabeled HeLa-TDS cells | Other | | High-Throughput Technology Development Studio (MPI-CBG, Dresden); |
| Cell line (*Homo sapiens*) | HeLa-TDS cells, stably expressing pEGFP-α-tubulin | Used in our previous work (*Kajtez et al., 2016*) | | |
| Cell line (*Homo sapiens*) | HeLa cells stably expressing YFP-tubulin | Gift from Lars Jansen | | |
| Cell line (*Homo sapiens*) | HeLa cells permanently transfected with EGFP-CLASP1 | Gift from Helder Maiato | | |

*Continued on next page*



*Continued*

| Reagent type (species) or resource | Designation | Source or reference | Identifiers | Additional information |
|---|---|---|---|---|
| Transfected construct (human) | PRC1 siRNA | Dharmacon | *Accell* A-019491-15-0020 | |
| Transfected construct (human) | Non targeting; mock siRNA | Dharmacon | *Accell Non-targeting Pool* D-001910-10-05 | |
| Transfected construct (human) | Kif4A siRNA | Santa Cruz Biotechnologies | sc-60888 | |
| Transfected construct (human) | Kif18A siRNA | Ambion | *Silencer Select Validated* Kif18A siRNA (s37882) | |
| Antibody | Anti-PRC1 (mouse monoclonal) | Santa Cruz Biotechnology | C-1; sc-376983 | IF (1:100) |
| Antibody | Anti-α-tubulin (rabbit polyclonal) | Sigma-Aldrich Corporation | RRID:AB_10743646 SAB4500087 | IF (1:100) |
| Antibody | Anti-Kif4A (mouse monoclonal) | Santa Cruz Biotechnology | RRID:AB_10707683 E-8; sc-365144 | IF (1:100) |
| Antibody | Anti-MKLP1 (rabbit polyclonal) | Santa Cruz Biotechnology | RRID:AB_631959 N-19; sc-867 | IF (1:100) |
| Antibody | Anti-Eg5 (mouse monoclonal) | Santa Cruz Biotechnology | RRID:AB_10841907 A-1; sc-365681 | IF (1:100) |
| Antibody | Anti-mouse IgG Alexa Fluor 488 (donkey polyclonal) | Abcam | ab150109 | IF (1:250) |
| Antibody | Anti-rabbit IgG Alexa Fluor 594 (donkey polyclonal) | Abcam | ab150064 | IF (1:250) |
| Antibody | Anti-rabbit IgG Alexa Fluor 405 (donkey polyclonal) | Abcam | RRID:AB_2715515 ab175649 | IF (1:250) |
| Antibody | Anti-mouse IgG Alexa Fluor 647 (goat polyclonal) | Abcam | RRID:AB_2811129 ab150119 | IF (1:250) |
| Antibody | Anti-Kif4A (rabbit polyclonal) | Bethyl | RRID:AB_2280904 A301-074A | IF (1:200) |
| Antibody | Anti-Kif18A (rabbit polyclonal) | Bethyl | RRID:AB_2296551 A301-080A | IF (1:100) |
| Recombinant DNA reagent | PRC1-tgRFPt-SspB WT (opto-PRC1) plasmid | This paper | | |
| Recombinant DNA reagent | His6-PRC1 plasmid | Addgene (*Nixon et al., 2015*) | RRID:Addgene_69111 | |
| Recombinant DNA reagent | tgRFPt-SspB WT plasmid | Addgene (*Guntas et al., 2015*) | RRID:Addgene_60415 | |
| Recombinant DNA reagent | iLID-CAAX plasmid | Addgene (*O'Neill et al., 2016*) | RRID:Addgene_85680 | |
| Recombinant DNA reagent | pEGFP-C1Kif4a-sires plasmid | Gift from Jason Stumpff | | |

*Continued on next page*

*Continued*

| Reagent type (species) or resource | Designation | Source or reference | Identifiers | Additional information |
|---|---|---|---|---|
| Recombinant DNA reagent | EGFP-Kif18A plasmid | Gift from Jason Stumpff | | |
| Recombinant DNA reagent | GFP-CENP-E plasmid | Gift from Marin Barišić | | |
| Recombinant DNA reagent | mRFP-CENP-B plasmid | Gift from Linda Wordeman | RRID:Addgene_23006 pMX234 | |
| Sequence-based reagent | FWD primer from His6-PRC1 plasmid | This paper | | GCTAGAATT GACCGGATG AGGAGAAGT GAGGTGCTG |
| Sequence-based reagent | REV primer from His6-PRC1 plasmid | This paper | | CATGGTGGC GACCGGTAA ATTCGAAGC TTGAGCTCG AGATCTGA GGGACTGG ATGTTGGT TGAATTGAGG |
| Commercial assay or kit | MycoAlert Mycoplasma Detection Kit | Lonza | #: LT07-118 | |
| Commercial assay or kit | In-Fusion HD Cloning Kit | Clontech | | |
| Commercial assay or kit | Nucleofector Kit | Lonza | #VVCA-1001 | Used with the Nucleofector 2b Device |
| Software, algorithm | RStudio | R Foundation for Statistical Computing | RRID:SCR_000432 | |
| Software, algorithm | MATLAB | MathWorks | RRID:SCR_001622 | |
| Software, algorithm | ImageJ | National Institutes of Health | RRID:SCR_003070 | |
| Software, algorithm | Adobe Illustrator CS5 | Adobe Systems | RRID:SCR_010279 | |
| Software, algorithm | *Low Light Tracking Tool* (LLTT) | ImageJ | | ImageJ plugin |
| Chemical compound, drug | DAPI stain | Sigma-Aldrich | D9542 | (1 µg/mL) |
| Chemical compound, drug | SiR-DNA | Spirochrome AG | #SC007 | 150 nM |
| Chemical compound, drug | SiR-tubulin | Spirochrome AG | #SC002 | 50 nM |
| Chemical compound, drug | Verapamil | Spirochrome AG | | 1 µM |
| Chemical compound, drug | Proteasome inhibitor MG-132 | Sigma-Aldrich | M8699 | 20 µM |

*Continued on next page*

*Continued*

| Reagent type (species) or resource | Designation | Source or reference | Identifiers | Additional information |
|---|---|---|---|---|
| Chemical compound, drug | Normal goat serum (NGS) | Sigma-Aldrich | 566380 | 1% for primary antibody solution, 2% for secondary antibody solution |
| Chemical compound, drug | Geneticin | Life technologies | Cat# 11811031 | |
| Chemical compound, drug | Penicillin/ streptomycin solution | Lonza | Cat# DE17-502E | |

## Cell lines

Experiments were performed using: unlabeled human osteosarcoma U2OS cell line, U2OS cells expressing CENP-A-GFP, mCherry-α-tubulin and PA-GFP-tubulin and U2OS cell line stably expressing CENP-A-GFP, used in our previous work (*Vukušić et al., 2017*), which were a gift from Marin Barišić and Helder Maiato (Institute for Molecular Cell Biology, University of Porto, Portugal); U2OS cell line stably expressing 2xGFP-EB3 and mCherry-CENP-A, a gift from Julie Welburn (University of Edinburgh, United Kingdom) (*Kajtez et al., 2016*); HeLa-Kyoto BAC lines stably expressing MKLP1-GFP, Kif4A-GFP, Kif18A-GFP, CENP-E-GFP and PRC1-GFP were a courtesy of Ina Poser and Tony Hyman (MPI-CBG, Dresden, Germany); unlabeled HeLa-TDS cells from the High-Throughput Technology Development Studio (MPI-CBG, Dresden); HeLa-TDS cells, permanently transfected with pEGFP-α-tubulin, used in our previous work (*Kajtez et al., 2016*); HeLa cells stably expressing YFP-tubulin, a courtesy of Lars Jansen (University of Oxford, United Kingdom); HeLa cells permanently transfected with EGFP-CLASP1, which was a gift from Helder Maiato (Institute for Molecular Cell Biology, University of Porto, Portugal). Cells were grown in flasks in Dulbecco's Modified Eagle's Medium (DMEM; Lonza, Basel, Switzerland) with 1 g/L D-glucose, L-glutamine, and pyruvate, supplemented with 10% of heat-inactivated Fetal Bovine Serum (FBS; Sigma Aldrich, St. Louis, MO, USA), 100 IU/mL penicillin and 100 mg/mL streptomycin solution (Lonza). For cell lines with stable expression of fluorescently labeled proteins, 50 µg/mL geneticin (Life Technologies, Waltham, MA, USA) was added. Cells were kept at 37°C and 5% $CO_2$ in a Galaxy 170 R humidified incubator (Eppendorf, Hamburg, Germany). All used cell lines were confirmed to be mycoplasma free by using MycoAlert Mycoplasma Detection Kit (Lonza).

## Plasmids

To make PRC1-tgRFPt-SspB WT (opto-PRC1) plasmid, PRC1 fragment was amplified from His6-PRC1 plasmid (RRID:Addgene_69111) (*Nixon et al., 2015*) using the primers GCTAGAA TTGACCGGATGAGGAGAAGTGAGGTGCTG (FWD) and CATGGTGGCGACCGGTAAATTCGAAGC TTGAGCTCGAGATCTGAGGGACTGGATGTTGGTTGAATTGAGG (REV) and inserted into plasmid tgRFPt-SspB WT (RRID:Addgene_60415) (*Guntas et al., 2015*) using *AgeI* restriction site. This step was performed using commercially available *In-Fusion HD Cloning Kit* (Clontech, Mountain View, CA, USA). The produced plasmid expresses PRC1 tagged with tgRFPt and SspB at the C-terminus. Plasmid iLID-CAAX was purchased (RRID:Addgene_85680) (*O'Neill et al., 2016*). Plasmids pEGFP-C1Kif4a-sires and EGFP-Kif18A were a gift from Jason Stumpff (University of Vermont, Burlington, VT, USA) (*Stumpff et al., 2008*; *Stumpff et al., 2012*). Plasmid GFP-CENP-E was a gift from Marin Barišić (Danish Cancer Society Research Center, Copenhagen, Denmark). Plasmid mRFP-CENP-B (pMX234; RRID:Addgene_23006) was provided by Linda Wordeman (University of Washington).

## Sample preparation

For depletion of endogenous PRC1 before opto experiments, cells were transfected 72 hr (U2OS cells) or 24 hr (HeLa cells) prior to imaging with 25 µL of 20 µM *Accell* PRC1 siRNA (A-019491-15-0020, Dharmacon, Lafayette, CO, USA) targeting 3' UTR of PRC1 mRNA. A day prior to imaging, siRNA-treated cells were transfected with corresponding plasmids in following amounts: 0.3 µg of

iLID-CAAX, 5.5 µg PRC1-tgRFPt-SspB-WT (resistant to the used RNAi), 0.5 µg pEGFP-C1Kif4a-sires, 1 µg EGFP-Kif18A, 1 µg GFP-CENP-E, and 2.5 µg mRFP-CENP-B. In HeLa BAC lines, 24 hr prior to imaging, mock experiment cells were transfected with 100 nM *Accell Non-targeting Pool* (D-001910-10-05; Dharmacon), PRC1 siRNA treated with 100 nM *Accell* PRC1 siRNA (Dharmacon), Kif4A siRNA treated with 100 nM Kif4A siRNA (sc-60888; Santa Cruz Biotechnologies, Dallas, TX, USA), whereas Kif18A siRNA treated with 100 nM *Silencer Select Validated* Kif18A siRNA (s37882; Ambion, Austin, TX, USA). All transfections were performed using Nucleofector Kit R with the Nucleofector 2b Device (Lonza) using X-001 program for U2OS and O-005 (high viability) program for HeLa cells. Following transfection, the cells were seeded on 35 mm glass coverslip uncoated dishes with 0.17 mm (1.5 coverglass) glass thickness (MatTek Corporation, Ashland, MA, USA) in 1.5 mL DMEM medium with appropriate supplements.

To visualize microtubules, cells were stained with silicon rhodamine (SiR)-tubulin (*Lukinavičius et al., 2014*; Spirochrome AG, Stein am Rhein, Switzerland), a far-red tubulin dye, at a concentration of 50 nM 12–16 hr prior to imaging. To prevent dye efflux, verapamil, a broad-spectrum efflux-pump inhibitor (Spirochrome Ltd.), was added in U2OS cells at a concentration of 1 µM. To visualize chromosomes and determine the phase of the mitosis, 20 min prior to imaging SiR-DNA (*Lukinavičius et al., 2015*; Spirochrome AG, Stein am Rhein, Switzerland) was added to a final concentration of 150 nM. For experiments on U2OS cells expressing 2xGFP-EB3 and mCherry-CENP-A, the cells were synchronized by adding 20 µM of the proteasome inhibitor MG-132 (Sigma-Aldrich) to arrest the cells in metaphase. Imaging was started 15 min after adding MG-132.

## Immunocytochemistry

Cells were fixed with ice-cold methanol for 3 min, washed three times with phosphate buffer saline (PBS), followed by 15 min permeabilization with 0.5% Triton in PBS. Cells were washed three times with PBS and blocked in 1% Normal Goat Serum (NGS) except in experiment for intensity of opto-PRC1 where cells were blocked in BSA in PBS for 1 hr at 4˚C. Cells were washed three times with PBS and then incubated in primary antibody solution in blocking buffer over night at 4˚C. Following primary antibodies were used: mouse anti-PRC1 monoclonal antibody (1:100; C-1; sc-376983, Santa Cruz Biotechnology), rabbit anti-α-tubulin polyclonal antibody (1:100; RRID:AB_10743646; SAB4500087; Sigma-Aldrich Corporation, St. Louis, MO, USA), mouse monoclonal anti-Kif4A antibody (1:100; RRID:AB_10707683; E-8; sc-365144; Santa Cruz Biotechnology), rabbit polyclonal anti-MKLP1 antibody (1:100; RRID:AB_631959; N-19; sc-867, Santa Cruz Biotechnology), mouse monoclonal anti-Eg5 antibody (1:100; RRID:AB_10841907; A-1; sc-365681, Santa Cruz Biotechnology), rabbit polyclonal anti-Kif18A antibody (1:100; RRID:AB_2296551; A301-080A; Bethyl), rabbit polyclonal anti-Kif4A antibody (1:200; RRID:AB_2280904; A301-074A; Bethyl). After washing off primary antibodies with PBS, cells were incubated in a solution of secondary antibodies in 2% NGS or BSA in PBS for 1 hr at room temperature protected from light. Following secondary antibodies were used: donkey anti-mouse IgG Alexa Fluor 488 (1:250; ab150109, Abcam, Cambridge, UK), donkey anti-rabbit IgG Alexa Fluor 594 (1:250; ab150064, Abcam), donkey anti-rabbit IgG Alexa Fluor 405 (1:250; RRID:AB_2715515; ab175649, Abcam), and goat anti-mouse IgG Alexa Fluor 647 (1:250; RRID:AB_2811129; ab150119, Abcam). After washing off the secondary antibodies three times in PBS, cells were incubated with a solution of 4',6-diamidino-2-phenylindole (DAPI) (1:1000) in PBS for 20 min and washed three times in PBS or SiR-DNA (150 nM) in PBS for 15 min before imaging. Note that we used immunocytochemistry for PRC1 rather than Western blot analysis because the efficiency of opto-PRC1 plasmid transfection is low, and as Western blot analysis provides information about the complete cell population, these results may not be relevant for the cells used in the opto-genetic experiments. In contrast, by using immunocytochemistry we analyzed only the cells with a similar opto-PRC1 level and in the same phase as those in our optogenetic experiments.

## Microscopy

Immunocytochemistry imaging and live imaging of unlabeled U2OS, U2OS stably expressing CENPA-GFP, HeLa-TDS pEGFP-α-tubulin and HeLa BAC CENP-E-GFP cells was performed using Bruker Opterra Multipoint Scanning Confocal Microscope (Bruker Nano Surfaces, Middleton, WI, USA), described previously (*Buđa et al., 2017*). In brief, the system was mounted on a Nikon Ti-E inverted microscope equipped with a Nikon CFI Plan Apo VC 100x/1.4 numerical aperture oil

objective (Nikon, Tokyo, Japan). During imaging, live cells were maintained at 37°C using H301-K-frame heating chamber (Okolab, Pozzuoli, NA, Italy). In order to obtain the optimal balance between spatial resolution and signal-to-noise ratio, 60 µm pinhole aperture was used. Opterra Dichroic and Barrier Filter Set 405/488/561/640 was used to separate the excitation light from the emitted fluorescence. Following emission filters were used: BL HC 525/30, BL HC 600/37, and BL HC 673/11 (all from Semrock, Rochester, NY, USA). Images were captured with an Evolve 512 Delta Electron Multiplying Charge Coupled Device (EMCCD) Camera (Photometrics, Tucson, AZ, USA) using a 200 ms exposure time. Electron multiplying gain was set on 400. Camera readout mode was 20 MHz. No binning was performed. The xy-pixel size in the image was 83 nm. The system was controlled with the Prairie View Imaging Software (Bruker).

For kinetics experiments on U2OS cells (*Figure 1*), 561 and 488 nm diode laser lines were used every 10 s with 200 ms exposure time. In all other optogenetic experiments, stacks were acquired using sequentially the following diode laser lines: 561 nm (to visualize opto-PRC1), 488 nm (to activate the optogenetic system and to visualize GFP), and 640 nm (to visualize SiR-tubulin or SiR-DNA, when applicable), with time interval between z-stacks of 60 s and with 200 ms exposure time per laser line. To prevent dissociation of PRC1 from the cell membrane between acquiring two consecutive z-stacks, only blue light was turned on for 200 ms every 10 s. Cells were imaged this way for 20 min in order to achieve almost complete removal of PRC1 from the spindle, after which the blue light was turned off and imaging was continued for another 10 min at 60 s intervals. The total imaging time of 30 min was chosen to be close to the typical metaphase duration of 29.7 ± 2.3 min, which was measured from the metaphase plate formation until anaphase onset in U2OS cells expressing CENP-A-GFP, mCherry-α-tubulin and PA-GFP-tubulin (N = 187) imaged after nuclear envelope breakdown every minute by obtaining 15 z-slices with 0.5 µm spacing and 150 ms exposure time. After 30 min of imaging, one z-stack in each of the three channels was taken in order to visualize the spindle and kinetochores after PRC1 return. In all cells except HeLa cells expressing pEGFP-α-tubulin, three focal planes with spacing between adjacent planes of 1 µm were acquired. Live imaging of HeLa cells stably expressing pEGFP-α-tubulin for measurements of tubulin intensities after acute PRC1 removal was performed in the same manner as described above for optogenetic experiments. Additionally, before turning the blue light on every 10 s, one z-stack was acquired using 561, 488, and 640 nm diode laser lines with averaging 8, and seven focal planes with spacing between adjacent planes of 0.5 µm. Stack was taken in the same way after 20 min of exposure to the blue light and again 10 min after the blue light was switched off. The same HeLa cells were used for measurements of tubulin intensities after long-term PRC1 removal and imaged by acquiring one z-stack using 561, 488, and 640 nm diode laser lines with averaging 8, and seven focal planes with spacing between adjacent planes of 0.5 µm. Imaging of HeLa BAC CENP-E-GFP mock and PRC1 siRNA-treated cells was performed by acquiring one z-stack of 3 focal planes with spacing between adjacent planes of 1 µm. For imaging of immunostained cells, five focal planes with spacing between adjacent planes of 0.5 µm were acquired.

Live imaging of U2OS cells stably expressing 2x-GFP-EB3 and mCherry-CENP-A and U2OS cells with transient expression of opto-PRC1, iLID-CAAX and GFP-Kif4A or EGFP-Kif18A was performed on a spinning disk confocal microscope system (Dragonfly, Andor Technology, Belfast, UK) using 63x/1.47 HC PL APO glycerol objective (Leica) and Zyla 4.2P scientific complementary metal oxide semiconductor (sCMOS) camera (Andor Technologies). During imaging cells were maintained at 37° and 5% CO2 within H301-T heating chamber (Okolab, Pozzuoli, Italy). Images were acquired using Fusion software (v 2.2.0.38). For live imaging of U2OS cells expressing 2x-GFP-EB3, mCherry-CENP-A and opto-PRC1, 488 nm and 561 nm laser lines were used for excitation to visualize GFP, and mCherry and opto-PRC1, respectively. In order to achieve PRC1 removal from the spindle, 3 z-planes with a z-spacing of 1 µm were acquired sequentially with both laser lines, every 10 s with 200 ms exposure time for 10 min. This imaging protocol was followed by five minutes of faster imaging, every 1.5 s with both laser lines on a central z-plane in order to visualize EB3 dynamics and to prevent the opto-PRC1 return. Control, mock and PRC1 siRNA-treated cells were imaged with the same imaging protocol. Live imaging of U2OS cells with transient expression of opto-PRC1, iLID-CAAX and GFP-Kif4A or EGFP-Kif18A was performed in the similar manner as described above. Additionally, to better visualize the localization of the Kif4A and Kif18A, before turning the blue light on every 10 s, one z-stack was acquired using 561, 488 and 640 nm diode laser lines with frame averaging 4, and three focal planes with spacing between adjacent planes of 1 µm. Stack was taken in

the same way after 12 min of exposure to the blue light and again 5 min after the blue light was switched off. Note that the exposure to the blue light was shorter than in previous experiments regarding opto-PRC1 kinetics so that the majority of the cells remained in metaphase during the experiment.

Live imaging of unlabeled, BAC (except CENP-E-GFP), YFP-tubulin, and EGFP-CLASP1 HeLa cell lines was performed on Leica TCS SP8 X laser scanning confocal microscope with a HC PL APO 63x/1.4 oil immersion objective (Leica, Wetzlar, Germany) heated with an objective integrated heater system (Okolab, Burlingame, CA, USA). During imaging, cells were maintained at 37°C in Okolab stage top heating chamber (Okolab, Burlingame, CA, USA). The system was controlled with the Leica Application Suite X software (LASX, 1.8.1.13759, Leica, Wetzlar, Germany). For GFP- and YFP-labeled proteins, a 488 nm and 514 nm Argon laser was used, respectively, and for SiR-DNA or SiR-tubulin, a 652 nm white light laser was used. For AF-647, a 637 nm white light laser was used. GFP, and SiR-DNA, SiR-tubulin or AF-647 emissions were detected with hybrid detector. For mock, PRC1 siRNA, Kif4A siRNA, and Kif18A siRNA experiments, images were acquired at 1–3 focal planes with 1 μm spacing and 0.05 μm pixel size. In optogenetic experiments 3 z-stacks with 1 μm spacing were acquired sequentially every 10 s in the same manner as in optogenetic experiments in U2OS cells. One z-stack with line averaging of 6 or 16 was acquired before system activation, 20 min after exposure to blue light and 10 min after the light was switched off.

## Image and data analysis

Since the cells were transiently transfected with opto-PRC1, we observed variability in PRC1 expression levels and therefore we imaged and analyzed only those metaphase spindles with PRC1 localization consistent with endogenous and fluorescently labeled PRC1 (*Kajtez et al., 2016*; *Polak et al., 2017*). Cells were not synchronized in order to avoid additional chemical treatment of cells, and metaphase was determined by alignment of kinetochores in the equatorial plane.

For determination of kinetics of PRC1 removal and return (*Figure 1C*), intensity of opto-PRC1 was measured in each time frame on one focal plane. We used *Polygon selection* tool in Fiji (National Institutes of Health, Bethesda, MD, USA) to encompass the area of the spindle, $A_{spindle}$, and measure mean spindle intensity, $M_{spindle}$. Mean background intensity in the cytoplasm was measured using $2.5 \times 2.5$ μm rectangle, $M_{cyto}$. Spindle intensity was background corrected by subtracting $M_{cyto}$ from $M_{spindle}$ to obtain $M_{spindle\ corr}$. In order to calculate the sum of PRC1 intensity on the spindle, $M_{spindle\ corr}$ was multiplied with $A_{spindle}$ for each timeframe. The background intensity outside of the cell was negligible, thus we did not take it into account. Note that for the measurements of kinetic parameters in *Figure 1C*, four outliers were excluded (see *Figure 1—figure supplement 1F*). The percentage of PRC1 removal was calculated from the mean value of intensity of all cells at time 20 min, that is, the last time point of the continuous exposure to the blue light. The percentage of return was calculated from the mean value of intensity of all cells in the interval 25–30 min, that is, during last 5 min of PRC1 return.

Intensity profiles of opto-PRC1 removal and return (*Figure 1D*) were obtained on sum intensity projections of all three z-planes by drawing a pole-to-pole line along the long axis of the spindle by using *Line* tool in Fiji. The width of the line corresponded to the width of each individual spindle. Intensities were normalized to position of the poles.

For quantification of PRC1 knock-down by siRNA and intensity level of opto-PRC1 (in *Figure 1—figure supplement 1A,B*) PRC1 intensity on fixed cells was measured on a sum-intensity projection of five focal planes by the procedure described above, in a way that mean spindle intensity was background corrected by subtracting mean intensity in the cytoplasm. For measuring opto-PRC1 intensity on the spindle, cells where PRC1 was visible on astral microtubules were not analyzed, nor imaged in live experiments. For quantification of PRC1 knock-down by siRNA in HeLa BAC MKLP1-GFP cell line, PRC1 intensity was quantified in the same manner as in U2OS cells.

Inter-kinetochore distance was measured using *Line* tool in Fiji on individual or maximum intensity z-projections of up to two z-planes as a distance between centers of sister kinetochore signals. Peripheral kinetochores were defined as three outermost pairs on each side of the spindle with respect to spindle long axis, while the remaining were considered as central. Measurement of inter-kinetochore distances in prometaphase was performed on U2OS cells expressing CENP-A-GFP, mCherry-α-tubulin and PA-GFP-tubulin, just after nuclear envelope breakdown in one imaging plane where both sister kinetochores could be clearly distinguished. For measuring kinetochore alignment

and orientation, as well as orientation and length of PRC1 bundles, *Multipoint* tool in Fiji was used. A point was placed in the center of signal of each sister kinetochore or edges of PRC1 signal for each bundle. Before measuring, images were rotated in order to achieve perpendicular direction of the equatorial plane with respect to x-axis. The equatorial plane was defined with two points placed between outermost pairs of kinetochores on the opposite sides of the spindle. For all measurements regarding kinetochores, those located at the spindle poles were not taken into account. All measurements in opto and control cells were performed in three time-points: before the blue light was switched on, after 20 min of continuous exposure to the blue light, and 10 min after the blue light was turned off. In untreated and PRC1 siRNA-treated cells measurements were performed at the beginning of imaging. Kymographs of kinetochore oscillations were produced by *Low Light Tracking Tool* (LLTT), an ImageJ plugin (*Krull et al., 2014*). Tracking of kinetochores in x, y plane was performed on maximum intensity of 2–3 z-planes. Sigma value (standard deviation of the Gaussian used to approximate the Point Spread Function (PSF) of the tracked objects) was set to *Automatic*.

EB3 spots in U2OS cells stably expressing 2x-GFP-EB3 and mCherry-CENP-A were tracked by obtaining their xy coordinates using *Multipoint* tool in Fiji from the frame when a spot appeared until it disappeared or was no longer clearly distinguishable from its neighbors. Only spots belonging to bridging fibers were traced, which were defined as those passing between sister kinetochores or moving along PRC1 streaks. Half-overlap length was measured as the distance between the last location where a tracked EB3 spot was visible and the spindle equator. The number of spots in time was obtained by visually inspecting time frames where individual kinetochore pair or PRC1 bundle was visible, and dividing the total number of the observed EB3 spots in the bridging fibers by the total time of observation of individual kinetochore pairs or PRC1 bundles. Kymographs were generated using *KymographBuilder* plugin in Fiji, and half-overlap length was measured from kymographs as the distance between a spot's trajectory end point and the mid-line between the poles corresponding to the equatorial plane. These measurements were performed in the first 2 min of a fast imaging sequence, which followed after the 10 min imaging protocol required for PRC1 removal.

The tubulin-GFP signal intensity of a cross-section of a bridging fiber was measured by drawing a 3-pixel-thick line between and perpendicular to the tubulin signal joining the tips of sister k-fibers. The same method was used in cells containing a kinetochore marker, that is, the profile intensity of the bridging fiber was extracted by looking at the tubulin channel and drawing the line between tips of k-fibers. For confirmation, we subsequently checked the kinetochore channel, and observed that this line was always placed between sister kinetochores. This validates our measurements of bridging fibers in opto cells using tubulin-GFP only. The tubulin intensity profile was corrected by subtracting mean background signal present in the cytoplasm (see *Figure 4A* and *Figure 4—figure supplement 1C,G*). The signal intensity of the bridging fiber was calculated as the area under the peak using Sci-Davis (Free Software Foundation Inc, Boston, MA, USA) (*Figure 4E*). The signal intensity in the region lateral from k-fiber tip was measured in a similar manner, 0.882 ± 0.04 µm away from the k-fiber tip (*Figure 4—figure supplement 1C,G*). This intensity corresponds to the sum of the bridging and k-fiber, given that PRC1 overlap half-length is longer and EB3 spots pass further than the position where the intensity profiles were measured. Therefore, the intensity of the k-fiber was calculated by subtracting the bridging fiber intensity from the corresponding sum of bridging and k-fiber intensity. The profile intensity of bridging fiber and at the position lateral from k-fiber tip in the cells stained with SiR-tubulin (*Figure 4B,C*) was performed in the same manner. All the measurements were performed on a single z-plane. Note that mostly outermost fibers were used for these measurements because of being most easily distinguished from neighboring fibers. All measurements in opto and control cells were performed at three time-points: before the blue light was switched on, after 20 min of continuous exposure to the blue light, and 10 min after the blue light was turned off.

Shapes of the spindle were quantified by tracking outermost k-fiber contours in the central z-slice of the spindle, or maximum intensity z-projection of two central z-slices. All spindles were rotated to have horizontal long axis. Pole-to-k-fiber-tip tracking was done using *Multipoint* tool by placing five roughly equidistant points along contour length, first point being at the pole and the last point being at the k-fiber tip. First point of each contour was translated to (0,0). This was done for maximum of 4 trackable outermost k-fibers in the spindle. Curvature of the contour was calculated by fitting a circle to the contour points of individual k-fibers and retrieving reciprocal value of its radius. To test the effect of tracking errors on curvature, we introduced 1-pixel noise to the x and y values

of tracked points, which did not change the result. Angle between outermost k-fibers (θ) was calculated as the angle between lines passing through last two points along the contour of sister k-fibers.

Spindle length and width were measured using *Line* tool in Fiji. For length measurements, a line was drawn from pole to pole. The position of the pole was defined as the location of the strongest tubulin signal. Width was measured by drawing a line between outermost kinetochore pairs on the opposite sides of the spindle, perpendicular to the spindle long axis.

To avoid the signal of Kif4A at the chromosomes, Kif4A bridging fiber intensity was measured on individual z-planes with $1 \times 0.5$ µm rectangles at the positions lateral from sister kinetochores and chromosomes, where no DNA (DAPI or SiR-DNA) was found, yet lateral parts of PRC1 streaks are present. This approach was applied in cases when PRC1 was not depleted or relocated to the cell membrane, that is, in opto cells before acute PRC1 removal and untreated cells. When PRC1 was not present on the spindle, that is, in opto cells after acute PRC1 removal and in PRC1 siRNA-treated cells, we placed rectangles at positions lateral from kinetochores and chromosomes, where antiparallel zones are thought to be positioned, considering their location before removal. The rectangles were placed parallel to the overlap. Given that bridging fiber intensity was measured mostly in the inner parts of the spindle, mean values from the spindle poles were considered as the Kif4A background and subtracted from Kif4A intensity retrieved from the positions of bridging fibers. For the measurements of Kif4A in the bridging fiber, we chose only cells in which Kif4A showed previously known localization on chromosomes and at comparable levels. This was important as immunostaining protocol sometimes resulted in cells with no or low chromosome staining, yet high spindle staining, and HeLa BAC Kif4A-GFP cells vary in expression levels among cells.

The intensity of Kif4A on chromosome arms in metaphase was measured in Kif4A channel on sum-intensity projections of all z-planes (nine planes in immunostained, and three in opto and BAC cells) using *Polygon selection* tool in Fiji by encompassing chromosomes in DAPI or SiR-DNA channel. In order to avoid the Kif4A signal on the spindle microtubules, the signal was measured only on the parts of the chromosome arms protruding into cytoplasm away from the spindle. For each cell, mean value of Kif4A arm intensity was calculated and corrected for background cytoplasm intensity measured in $2.5 \times 2.5$ µm rectangle by subtracting it from mean intensity of Kif4A. In immunostained cells only those with Kif4A localized on chromosomes were taken into account as in some sessions the Kif4A signal was very strong on the whole spindle and not present on the chromosomes both in untreated and PRC1 siRNA-treated cells.

The intensity of GFP-Kif4A on chromosomes in anaphase was measured in GFP-Kif4A channel on sum-intensity projections of all three z-planes using *Polygon selection* tool in Fiji by encompassing chromosomes in SiR-DNA channel. Background cytoplasm intensities in GFP-Kif4A channel were measured in $2.5 \times 2.5$ µm rectangle and subtracted from measured mean intensities of GFP-Kif4A. Corrected intensities were divided by the number of focal planes. In anaphase, measurements were performed 4 min after anaphase onset. Anaphase onset was defined as the timeframe when separation of majority of sister chromatids in SiR-DNA channel occurred.

The mean bridging fiber intensities of Kif18A in all treatments were obtained on a single z-plane using $0.4 \times 0.4$ µm rectangles in Fiji covering the Kif18A signal in the bridge. The rectangles of the same dimension were used to obtain the Kif18A signal from the sister k-fiber tips and the mean value for the pair was calculated. Since the measurements of Kif18A were mostly obtained on the outermost fibers, these values were corrected by subtracting mean background cytoplasm intensity measured in a $2.5 \times 2.5$ µm rectangle.

The intensity of MKLP1 in the bridging fibers was measured in GFP channel on a single z-plane using $0.4 \times 0.4$ µm rectangle in Fiji by placing it on each bridging fiber visible in the SiR-tubulin channel. These intensities were corrected for background GFP intensity in cytoplasm measured in $2.5 \times 2.5$ µm rectangle. For quantification of PRC1 knock-down by siRNA in HeLa BAC MKLP1-GFP cell line, PRC1 intensity was quantified in the same manner as in U2OS cells: on a sum-intensity projection of five focal planes in a way that mean spindle intensity was background corrected by subtracting mean intensity in the cytoplasm,.

The intensity of CENP-E and CLASP1 on plus ends-of k-fibers was measured using $0.4 \times 0.4$ µm rectangle encompassing plus-end of k-fibers on a single z-plane. Intensities were corrected for the intensity in cytoplasm measured using $2.5 \times 2.5$ µm rectangle in the central z-plane as this intensity was similar between different z-planes. For analysis of CLASP1 and CENP-E intensity on plus-ends of

k-fibers only cells with similar intensities on the spindle within each treatment were taken into account as there was a variation in expression level among cells.

Measurements regarding Kif4A, Kif18A, MKLP1, CENP-E, and CLASP1 were performed in two time-points in metaphase: before the blue light was switched on and at the end of continuous exposure to the blue light. The intensities were normalized on the mean value of control at each treatment, that is, before the blue light was switched on for acute PRC1 removal and untreated for long-term depletion.

Quantification of the signal length of PRC1-GFP in mock, Kif4A, and Kif18A siRNA-treated HeLa cells was performed by drawing a 5-pixel-thick pole-to-pole line along individual bundles and calculated as the width of the peak of the PRC1-GFP signal intensity.

Statistical analysis was performed in MATLAB (RRID:SCR_001622; MathWorks, Natick, MA, USA) and RStudio (RRID:SCR_000432; R Foundation for Statistical Computing, Vienna, Austria). Normality of the data was inspected by Q-Q plots and Shapiro-Wilk test of normality. Two groups with normally distributed data were tested with two-tailed t-test, while more than two groups were tested with one-way ANOVA followed by Tukey HSD post hoc test. Two groups with non-normally distributed data were tested with Mann-Whitney test, while more than two groups were tested with Kruskal-Wallis rank sum test followed by pairwise Wilcoxon rank sum test. Used statistical analysis is noted in figure captions. Proportions were statistically compared with test for equality of proportions, two-proportions z-test. For data with expected count smaller than 5, Yates's correction for continuity was used.

Graphs were generated in MATLAB (MathWorks) and RStudio (R Foundation for Statistical Computing). ImageJ (RRID:SCR_003070; National Institutes of Health, Bethesda, MD, USA) was used to crop and rotate images, and to adjust brightness and contrast to the entire image, which was applied equally to all images in the same panel. The images are rotated in order for the spindle to be horizontal in every time frame. To remove high-frequency noise in displayed images a Gaussian blur filter with a 0.5-pixel sigma (radius) was applied where stated. Figures were assembled in Adobe Illustrator CS5 (RRID:SCR_010279; Adobe Systems, Mountain View, CA, USA).

## Acknowledgements

We thank Helder Maiato and Marin Barišić for unlabeled U2OS and U2OS CENP-A-GFP cell lines; Julie Welburn for U2OS 2xGFP-EB3 mCherry-CENP-A cell line; Ina Poser and Tony Hyman for HeLa MKLP1-GFP, Kif4A-GFP, Kif18A-GFP, CENP-E-GFP and PRC1-GFP cell lines; Helder Maiato for HeLa EGFP-CLASP1 cell line; Lars Jansen for HeLa YFP-tubulin cell line; Marin Barišić for GFP-CENP-E plasmid; Jason Stumpff for pEGFP-C1Kif4a-sires and EGFP-Kif18A plasmids; Linda Wordeman for mRFP-CENP-B plasmid; Nenad Pavin, Agneza Bosilj, Kruno Vukušić, and Juraj Simunić for discussions and constructive comments on the manuscript; Sonja Lesjak for help with the cloning and plasmids; Ivana Šarić for the drawings. This work was funded by the European Research Council (ERC Consolidator Grant, GA number 647077, and ERC Synergy Grant, GA number 855158), Croatian Science Foundation (HRZZ, project IP-2014-09-4753), and the QuantiXLie Center of Excellence, a project co-financed by the Croatian Government and European Union through the European Regional Development Fund - the Competitiveness and Cohesion Operational Programme (Grant KK.01.1.1.01.0004).

## Additional information

### Funding

| Funder | Grant reference number | Author |
| --- | --- | --- |
| European Research Council | ERC,GA number 647077 | Iva M Tolić |
| Croatian Science Foundation | HRZZ project IP-2014-09-4753 | Iva M Tolić |
| European Research Council | ERC, GA number 855158 | Iva M Tolić |
| European Regional Development Fund | Grant KK.01.1.1.01.0004 | Iva M Tolić |

The funders had no role in study design, data collection and interpretation, or the decision to submit the work for publication.

## Author contributions
Mihaela Jagrić, Patrik Risteski, Software, Formal analysis, Validation, Investigation, Visualization, Methodology, Writing - original draft; Jelena Martinčić, Software, Formal analysis, Validation, Investigation, Visualization, Methodology; Ana Milas, Methodology; Iva M Tolić, Conceptualization, Supervision, Funding acquisition, Writing - review and editing

## Author ORCIDs
Mihaela Jagrić (iD) https://orcid.org/0000-0002-2296-0649
Patrik Risteski (iD) https://orcid.org/0000-0003-2137-7491
Jelena Martinčić (iD) https://orcid.org/0000-0001-5834-0938
Iva M Tolić (iD) https://orcid.org/0000-0003-1305-7922

## Decision letter and Author response
Decision letter https://doi.org/10.7554/eLife.61170.sa1
Author response https://doi.org/10.7554/eLife.61170.sa2

# Additional files

## Supplementary files
- Source code 1. Main codes used for data analysis.
- Supplementary file 1. Table of the data generated in this study, listed per figure.
- Transparent reporting form

## Data availability
All data generated or analysed during this study are included in the manuscript and supporting files.

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
