## [Decision Letter]

**Acceptance summary:**

This work reveals unknown functions of PRC1 in metaphase through the development of a fast optogenetic tool to remove PRC1 from bridging fibres within the spindle. This elegant work shows PRC1 contributes to chromosome alignment through the control of the length overlap of antiparallel fibres, and the number of fibres. The motors Kif4a and Kif18a also localize to the bridging fibres in a PRC1-dependent manner to contribute to balance forces at microtubule overlaps during chromosome alignment.

**Decision letter after peer review:**

Thank you for submitting your article "Optogenetic control of PRC1 reveals that bridging fibers promote chromosome alignment by overlap length-dependent forces" for consideration by *eLife*. Your article has been reviewed by four peer reviewers, one of whom is a member of our Board of Reviewing Editors, and the evaluation has been overseen by Anna Akhmanova as the Senior Editor. The reviewers have opted to remain anonymous.

The reviewers have discussed the reviews with one another and the Reviewing Editor has drafted this decision to help you prepare a revised submission.

Summary:

The authors establish an optogenetic methodology to remove PRC1 from overlapping fiber bridges within the spindle so that removal is acute. Upon illumination, PRC1 is relocalized to the cortex. The major conclusion from these studies is that bridging fibers have a previously undiscovered role in promoting chromosome alignment. The authors present an interesting model to explain this role that depends on kinesin-dependent control of overlap length within bridging fibers. They propose a longer overlap of bridging fibers leads to chromosome misalignment because of imbalanced forces generated by different length of bridging fibers. However, before this paper is published, more supporting data is required to support the role of kinesins in controlling the length of overlaps between k-fibers and bridging fibers.

Essential revisions:

1) This point was the major issue for the reviewers. They found the work and conclusions on the various kinesins and Clasp1 at bridging fibers not supported by enough data. The functional significance of these proteins on bridging fibers remains unclear. They suggested the authors focus their results on 1-2 kinesins, add supporting data and remove the weak data on CENP-E and Clasp1.

In addition, the authors claim that Kif4A, Kif18A, and MKLP1 all disappear from the bridging fiber during activation of opto-PRC1 and return when the cells are placed back in the dark. However, this is very unclear from the current presentation of the data, except perhaps for MKLP1. Kif4A in particular appears to still be on the bridging fiber during exposure, whereas Kif18A is difficult to see on the bridging fiber before exposure. The authors need to include insets demonstrating examples of this behavior as well as quantification (as they have done for most other data sets in the manuscript) to substantiate these claims. The N of these experiments (Table 1) is also quite low. Additionally, overlap lengths and chromosome alignment are not affected by PRC1 siRNA even though KIF4A and KIF18A are absent from bridging fibers. Why is that? Kif18a and Kif4a depletion leads to longer spindles. It is not clear whether the lengthening of PRC1-containing overlaps is a secondary effect of spindle length increases or is due to loss of kinesin-mediated regulation of bridging fiber length. This needs to be discussed as a possible outcome of PRC1 change in localization.

2) Discuss more the alternative to your model in the discussion: that PRC1's role in chromosome alignment may affect spindle dynamics rather than forces generated in the bridging fiber overlap, in the absence of data ruling out this possibility.

3) There are some concerns about the accuracy of measurements related to microtubule dynamics and numbers specifically within bridging fibers, especially after opto-PRC1 removal. The authors explain how they performed measurements of EB3 dynamics using kinetochores as markers. In Figure 3, they find that the number and behavior of dynamic microtubules within the bridging fibers are not changed following PRC1 removal. In contrast, the data in Figure 4 suggest that the total number of microtubules within bridging fibers is reduced by 2.5-fold. The authors conclude that the number of non-dynamic microtubules within bridging fibers are reduced by removal of PRC1. Kinetochore markers do not seem to be present in the cells used to measurement the microtubule number in individual bridging fibers, and it isn't clear how reliably these tubulin fluorescence measurements report bridging fiber thickness specifically. An independent test of this conclusion would be helpful. The methods for this assay should be better documented also to ensure independent reproducibility.

4) Edit the title to include mention of the spindle, to appeal to broader audience and dampen the conclusions about bridging fibers promoting chromosome alignment through overlap length-dependent forces. The data is consistent with the model and title but does not rule out PRC1 acting on microtubules and spindles rather than chromosome alignment directly.

The reviewers also pointed out they could not access the code used for data analysis. We encourage the authors to make code available as soon as possible to the community to promote open-source software and data/scripts access and reproducibility.

[Editors' note: further revisions were suggested prior to acceptance, as described below.]

Thank you for resubmitting your work entitled "Optogenetic control of PRC1 reveals its role in chromosome alignment on the spindle by overlap length-dependent forces" for further consideration by *eLife*. Your revised article has been evaluated by Anna Akhmanova (Senior Editor) and a Reviewing Editor.

The manuscript has been improved but there is one small remaining point that need to be addressed before acceptance, as outlined below:

While PRC1 distribution on overlaps changes in Kif18a RNA, it is not clear whether it is the consequence of spindle elongation linked to Kif18a RNAi – this was a comment we previously made. Instead of graph 5e (or additionally to), please show the ratio [overlap/spindle length] for control and each perturbation, with individual data points as a scatter plot.

---

## [Author Response]

Essential revisions:1) This point was the major issue for the reviewers. They found the work and conclusions on the various kinesins and Clasp1 at bridging fibers not supported by enough data. The functional significance of these proteins on bridging fibers remains unclear. They suggested the authors focus their results on 1-2 kinesins, add supporting data and remove the weak data on CENP-E and Clasp1.In addition, the authors claim that Kif4A, Kif18A, and MKLP1 all disappear from the bridging fiber during activation of opto-PRC1 and return when the cells are placed back in the dark. However, this is very unclear from the current presentation of the data, except perhaps for MKLP1. Kif4A in particular appears to still be on the bridging fiber during exposure, whereas Kif18A is difficult to see on the bridging fiber before exposure. The authors need to include insets demonstrating examples of this behavior as well as quantification (as they have done for most other data sets in the manuscript) to substantiate these claims. The N of these experiments (Table 1) is also quite low. Additionally, overlap lengths and chromosome alignment are not affected by PRC1 siRNA even though KIF4A and KIF18A are absent from bridging fibers. Why is that? Kif18a and Kif4a depletion leads to longer spindles. It is not clear whether the lengthening of PRC1-containing overlaps is a secondary effect of spindle length increases or is due to loss of kinesin-mediated regulation of bridging fiber length. This needs to be discussed as a possible outcome of PRC1 change in localization.

We thank the reviewers for raising these important points, which we addressed in the revised manuscript by new experiments and quantifications. In summary, we focused our results on Kif4A, Kif18A and MKLP1 in the bridging fiber, removed the data on CLASP1 and CENP-E in the bridging fiber and moved the data of their kinetochore localization to the supplement. We performed new sets of experiments for Kif4A and Kif18A to better demonstrate and quantify their localization in the bridging fibers and behavior upon acute and long-term PRC1 removal, and to increase the number of analyzed cells for most experiments. Importantly, we found that Kif4A signal in the bridging fibers decreased to a larger extent after optogenetic than after siRNA PRC1 depletion, suggesting that the strong reduction of Kif4A in the optogenetic experiment led to the enlargement of overlaps and chromosome misalignment, which was not observed in siRNA. These new results demonstrate a functional significance of Kif4A in the bridging fibers. On the other hand, we found that Kif18A signal in the bridging fibers decreased to a similar extent by the two methods of PRC1 depletion, and this decrease was similar to the decrease of the tubulin signal in the bridging fiber, suggesting that the amount of Kif18A per microtubule remained largely unaffected by PRC1 removal. Thus, Kif18A is not responsible for the change of overlap length after acute PRC1 removal, but its functional significance in the regulation of the overlap length of bridging microtubules, as well as the length of other spindle microtubules, is shown by Kif18A siRNA experiments. Our new results are incorporated in the revised manuscript as described below.

To demonstrate the presence of Kif4A and Kif18A in the bridging fiber, we performed new opto and immunostaining experiments and provided new figures containing multi-color images, individual black and white channels together with enlarged areas of the bridging fiber (Figure 5A,B; Figure 5—figure supplement 1A,B). We added the following text to the Results section: “Surprisingly, we found that Kif4A and Kif18A localize in the bridge during metaphase, visible as thin lines across or next to the location of sister kinetochores where PRC1-labelled bundles are found (Polak et al., 2017) (Figure 5A; Figure 5—figure supplement 1A). Similar localization was observed for MKLP1, spanning the region between two sister k-fibers (Figure 5A). In vertically oriented spindles, whose long axis was roughly perpendicular to the imaging plane, Kif4A and Kif18A colocalized with opto-PRC1, now visible as spots in the cross-section of the spindle (Figure 5—figure supplement 1B).”

Furthermore, we quantified the localization of Kif4A, Kif18A and MKLP1 in the bridging fibers before and after acute PRC1 removal as well as in untreated and PRC1 siRNA treated cells. These data are shown in new figures, graphs, schemes and the updated table (Figure 5A-D; Figure 5—figure supplement 1A-C; Table 1). The following text was added to the Results section:

“To explore whether PRC1 removal affects the localization of Kif4A, Kif18A and MKLP1, we analyzed their intensities within bridging fibers. Measurements of Kif4A intensity upon acute PRC1 removal and long term-depletion, in regions lateral from chromosomes which correspond to peripheral parts of bridging fibers, showed that its intensity in the bridging fibers decreased (Figure 5A-C; Figure 5—figure supplement 1A). Additional analysis of Kif4A intensity in HeLa cells after long-term PRC1 removal corroborated this result (Figure 5—figure supplement 1C). Interestingly, there was a larger reduction in Kif4A intensity after acute PRC1 removal, where it decreased by 76 ± 5%, in comparison to long-term depletion where it decreased by 52 ± 3% (p=1x10^-4^; Figure 5C; Table 1). Kif18A intensity in the bridging fibers also decreased upon acute PRC1 removal and long term-depletion (Figure 5A-C; Figure 5—figure supplement 1A,C). Yet, in contrast to Kif4A, Kif18A intensities in the bridging fiber decreased to a similar extent, i.e. by 43 ± 11 % and 38 ± 6 %, after acute and long-term PRC1 depletion, respectively (p=0.68; Figure 5C; Table 1). MKLP1 intensities were also similarly reduced after both approaches, 87 ± 5 % after acute PRC1 removal and 83 ± 11 % after long-term depletion (p=0.68; Figure 5D; Figure 5—figure supplement 1C).”

Regarding the differences in effects of acute and long-term PRC1 removal on kinetochore alignment and overlap lengths, and the effect of Kif4A and Kif18A depletions on spindle length, we added the following in the Results section:

“As Kif4A and Kif18A are known to regulate microtubule length (Mayr et al., 2007; Varga et al., 2006; Wandke et al., 2012), we set out to explore their roles in overlap length regulation. We hypothesized that these kinesins may regulate the length of bridging microtubules and hence their antiparallel overlaps during metaphase. Indeed, removal of Kif4A or Kif18A by siRNA resulted in longer PRC1-labeled overlaps. Kif4A siRNA treatment increased overlap length for 0.9 ± 0.2 µm, which was similar to spindle length increase of 1.1 ± 0.2 µm (p=0.66, t-test; Figure 5E). However, Kif18A siRNA treatment increased overlap length for 1.8 ± 0.3 µm, while the spindle length increased to a larger extent, for 3.0 ± 0.5 µm (p=0.03, t-test; Figure 5E). These results suggest that both Kif4A and Kif18A regulate the length of bridging microtubules, but Kif18A has a larger effect on other spindle microtubules than Kif4A.”

We also added the following text in the Discussion: “We propose that both Kif4A and Kif18A regulate the length of bridging microtubules and their overlaps under normal conditions (Figure 6C), as their depletion results in longer overlaps. Intriguingly, we found that Kif4A signal in the bridging fibers decreased to a larger extent after acute than after long-term PRC1 depletion, whereas Kif18A signal decreased to a similar extent by the two methods of PRC1 depletion. Moreover, Kif4A depletion by siRNA resulted in an increase of ~1 µm in overlap length, which was similar to the increase observed after acute PRC1 removal, whereas Kif18A depletion by siRNA led to a larger overlap increase. Thus, we suggest that the observed overlap elongation after acute PRC1 removal is a consequence of the strong reduction of Kif4A in the bridging fibers. In contrast, after long-term PRC1 depletion, we speculate that a larger fraction of Kif4A is present on the microtubules during overlap formation, contributing to the compensatory mechanism maintaining normal overlap length.”

Finally, all quantifications regarding localization of Kif4A, Kif18A and MKLP1 in the bridging fibers, Kif4A on the chromosome arms and Kif18A, CLASP1 and CENP-E on plus ends of k-fibers are thoroughly described and clarified in the Methods section: ”To avoid the signal of Kif4A at the chromosomes, Kif4A bridging fiber intensity was measured on individual z-planes with 1 x 0,5 µm rectangles at the positions lateral from sister kinetochores and chromosomes, where no DNA (DAPI or SiR-DNA) was found, yet lateral parts of PRC1 streaks are present. This approach was applied in cases when PRC1 was not depleted or relocated to the cell membrane, i.e., in opto cells before acute PRC1 removal and untreated cells. When PRC1 was not present on the spindle, i.e., in opto cells after acute PRC1 removal and in PRC1 siRNA treated cells, we placed rectangles at positions lateral from kinetochores and chromosomes, where antiparallel zones are thought to be positioned, considering their location before removal. The rectangles were placed parallel to the overlap. Given that bridging fiber intensity was measured mostly in the inner parts of the spindle, mean values from the spindle poles were considered as the Kif4A background and subtracted from Kif4A intensity retrieved from the positions of bridging fibers. For the measurements of Kif4A in the bridging fiber, we chose only cells in which Kif4A showed previously known localization on chromosomes and at comparable levels. This was important as immunostaining protocol sometimes resulted in cells with no or low chromosome staining, yet high spindle staining, and HeLa BAC Kif4A-GFP cells vary in expression levels among cells.

The intensity of Kif4A on chromosome arms in metaphase was measured in Kif4A channel on sum-intensity projections of all z-planes (nine planes in immunostained, and three in opto and BAC cells) using Polygon selection tool in Fiji by encompassing chromosomes in DAPI or SiR-DNA channel. In order to avoid the Kif4A signal on the spindle microtubules, the signal was measured only on the parts of the chromosome arms protruding into cytoplasm away from the spindle. For each cell mean value of Kif4A arm intensity was calculated and corrected for background cytoplasm intensity measured in 2.5 x 2.5 µm rectangle by subtracting it from mean intensity of Kif4A. In immunostained cells only those with Kif4A localized on chromosomes were taken into account as in some sessions the Kif4A signal was very strong on the whole spindle and not present on the chromosomes both in untreated and PRC1 siRNA treated cells.

The intensity of GFP-Kif4A on chromosomes in anaphase was measured in GFP-Kif4A channel on sum-intensity projections of all three z-planes using Polygon selection tool in Fiji by encompassing chromosomes in SiR-DNA channel. Background cytoplasm intensities in GFP-Kif4A channel was measured in 2.5 x 2.5 µm rectangle and subtracted from measured mean intensities of GFP-Kif4A. Corrected intensities were divided by the number of focal planes. In anaphase, measurements were performed 4 min after anaphase onset. Anaphase onset was defined as the timeframe when separation of majority of sister chromatids in SiR-DNA channel occurred.

The mean bridging fiber intensities of Kif18A in all treatments were obtained on a single z- plane using 0.4 x 0.4 µm rectangles in Fiji covering the Kif18A signal in the bridge. The rectangles of the same dimension were used to obtain the Kif18A signal from the sister k-fiber tips and the mean value for the pair was calculated. Since the measurements of Kif18A were mostly obtained on the outermost fibers, these values were corrected by subtracting mean background cytoplasm intensity measured in a 2.5 x 2.5 µm rectangle.

The intensity of MKLP1 in the bridging fibers was measured in GFP channel on a single z-plane using 0.4 x 0.4 µm rectangle in Fiji by placing it on each bridging fiber visible in the SiR-tubulin channel. These intensities were corrected for background GFP intensity in cytoplasm measured in 2.5 x 2.5 µm rectangle. For quantification of PRC1 knock-down by siRNA in HeLa BAC MKLP1-GFP cell line, PRC1 intensity was quantified in the same manner as in U2OS cells: on a sum-intensity projection of five focal planes in a way that mean spindle intensity was background corrected by subtracting mean intensity in the cytoplasm.

The intensity of CENP-E and CLASP1 on plus ends-of k-fibers was measured using 0.4 x 0.4 µm rectangle encompassing plus-end of k-fibers on a single z-plane. Intensities were corrected for the intensity in cytoplasm measured using 2.5 x 2.5 µm rectangle in the central z-plane as this intensity was similar between different z-planes. For analysis of CLASP1 and CENP-E intensity on plus-ends of k-fibers only cells with similar intensities on the spindle within each treatment were taken into account as there was a variation in expression level among cells.

Measurements regarding Kif4A, Kif18A, MKLP1, CENP-E and CLASP1 were performed in two time-points in metaphase: before the blue light was switched on and at the end of continuous exposure to the blue light. The intensities were normalized on the mean value of control at each treatment, i.e., before the blue light was switched on for acute PRC1 removal and untreated for long-term depletion.”

2) Discuss more the alternative to your model in the discussion: that PRC1's role in chromosome alignment may affect spindle dynamics rather than forces generated in the bridging fiber overlap, in the absence of data ruling out this possibility.

We agree with the suggestion of the reviewers. We added in the Discussion section the following: “The localization of Kif4A on chromosome arms, where it is involved in polar ejection forces (Bieling et al., 2010a; Brouhard and Hunt, 2005) was preserved after acute PRC1 removal. Similarly, Kif18A, CLASP1 and CENP-E remained on plus-ends of k-fibers. However, as we cannot exclude potential subtle changes in the localization of these proteins or mislocalization of other proteins, the observed chromosome misalignment and overlap elongation after acute PRC1 removal could also be promoted by the changes of the dynamics of other microtubules within the spindle consequently affecting forces produced by k-fibers and polar ejection forces.”

3) There are some concerns about the accuracy of measurements related to microtubule dynamics and numbers specifically within bridging fibers, especially after opto-PRC1 removal. The authors explain how they performed measurements of EB3 dynamics using kinetochores as markers. In Figure 3, they find that the number and behavior of dynamic microtubules within the bridging fibers are not changed following PRC1 removal. In contrast, the data in Figure 4 suggest that the total number of microtubules within bridging fibers is reduced by 2.5-fold. The authors conclude that the number of non-dynamic microtubules within bridging fibers are reduced by removal of PRC1. Kinetochore markers do not seem to be present in the cells used to measurement the microtubule number in individual bridging fibers, and it isn't clear how reliably these tubulin fluorescence measurements report bridging fiber thickness specifically. An independent test of this conclusion would be helpful. The methods for this assay should be better documented also to ensure independent reproducibility.

We thank the reviewers for this comment. To clarify how the tubulin intensities were measured in the HeLa tubulin-GFP line, we added multiple new example images where we indicated the starting and ending positions of lines drawn to obtain the signal intensities of the bridging fiber (Figure 4—figure supplement 1C). Moreover, we performed new experiments to measure bridging and k-fiber intensities on live untreated and PRC1 siRNA treated HeLa cells stably expressing tubulin-GFP where we added the kinetochore marker, mRFP-CENP-B. In those cells, the bridging fiber intensity was measured between sister kinetochores and new images demonstrating the intensity measurement assay are provided (Figure 4—figure supplement 1G-I).

The cell stained with SiR-tubulin (Figure 4B), where the intensity profile of the bridging fiber was acquired, also contains a kinetochore marker, CENP-A-GFP, but it was not previously shown in the images. In the revised manuscript, the channel containing the kinetochore marker is now added to Figure 4B, as well as the enlargements with the lines representing the positions where the bridging fiber was measured with respect to the position of kinetochores. We also included single color images of the enlarged area where the measurements were performed. The conclusion about the decrease of the tubulin signal in the bridging fiber obtained by using tubulin-GFP is supported by independent experiments based on SiR-tubulin.

In the Materials and methods section, we explained the assay in more detail and the control procedures to show that the tubulin fluorescence measurements reliably report bridging fiber thickness specifically:

“The tubulin-GFP signal intensity of a cross-section of a bridging fiber was measured by drawing a 3-pixel-thick line between and perpendicular to the tubulin signal joining the tips of sister k-fibers. […] All measurements in opto and control cells were performed at three time-points: before the blue light was switched on, after 20 min of continuous exposure to the blue light, and 10 min after the blue light was turned off.”

4) Edit the title to include mention of the spindle, to appeal to broader audience and dampen the conclusions about bridging fibers promoting chromosome alignment through overlap length-dependent forces. The data is consistent with the model and title but does not rule out PRC1 acting on microtubules and spindles rather than chromosome alignment directly.

The title is now changed to: “Optogenetic control of PRC1 reveals its role in chromosome alignment on the spindle by overlap length-dependent forces”.

The reviewers also pointed out they could not access the code used for data analysis. We encourage the authors to make code available as soon as possible to the community to promote open-source software and data/scripts access and reproducibility.

The codes used in the main analyses can be found as the Source code 1.

[Editors' note: further revisions were suggested prior to acceptance, as described below.]

The manuscript has been improved but there is one small remaining point that need to be addressed before acceptance, as outlined below:While PRC1 distribution on overlaps changes in Kif18a RNA, it is not clear whether it is the consequence of spindle elongation linked to Kif18a RNAi – this was a comment we previously made. Instead of graph 5e (or additionally to), please show the ratio [overlap/spindle length] for control and each perturbation, with individual data points as a scatter plot.

We added a graph showing the ratio of overlap and spindle length for control, Kif4A and Kif18A depletion, with individual data points as a scatter plot, in the new Figure 5F. The old graph showing the change in length of the overlap and spindle is now Figure 5G. We revised the text related to Figure 5:

"… Indeed, removal of Kif4A or Kif18A by siRNA resulted in longer PRC1-labeled overlaps and longer spindles (Figure 5E), with a small but not significant increase in the ratio of overlap length to spindle length (Figure 5F). However, the increase in the overlap and spindle length was significantly different after the two treatments (Figure 5G). Kif4A siRNA treatment increased overlap length for 0.9 ± 0.2 µm compared with control, which was similar to spindle length increase of 1.2 ± 0.3 µm (p=0.48, t-test; Figure 5G). In contrast, Kif18A siRNA treatment increased overlap length for 1.8 ± 0.3 µm, whereas the spindle length increased to a larger extent, for 3.7 ± 0.6 µm (p=0.004, t-test; Figure 5G). These results suggest that both Kif4A and Kif18A regulate the length of bridging microtubules and their overlap, but Kif18A has a greater effect on overall spindle microtubules and spindle length than Kif4A."